# State Combinatorial Generalization In Decision Making With Conditional Diffusion Models

**Xintong Duan, Yutong He, Fahim Tajwar, Wentse Chen, Ruslan Salakhutdinov, Jeff Schneider**
*Carnegie Mellon University*
*{xintongd,yutonghe,ftajwar,wentsec,rsalakhu,jeff4}@cs.cmu.edu*

**Reviewed on OpenReview:** *https://openreview.net/forum?id=XB1dd01Ozz*

## Abstract

Many real-world decision-making problems are combinatorial in nature, where states (e.g., surrounding traffic of a self-driving car) can be seen as a combination of basic elements (e.g., pedestrians, trees, and other cars). Due to combinatorial complexity, observing all combinations of basic elements in the training set is infeasible, which leads to an essential yet understudied problem of *zero-shot generalization to states that are unseen combinations of previously seen elements.* In this work, we first formalize this problem and then demonstrate how existing value-based reinforcement learning (RL) algorithms struggle due to unreliable value predictions in unseen states. We argue that this problem cannot be addressed with exploration alone, but requires more expressive and generalizable models. We demonstrate that behavior cloning with a conditioned diffusion model trained on successful trajectory generalizes better to states formed by new combinations of seen elements than traditional RL methods. Through experiments in maze, driving, and multiagent environments, we show that conditioned diffusion models outperform traditional RL techniques and highlight the broad applicability of our problem formulation.

## 1 Introduction

In many real-world decision-making tasks, environments can be broken down into combinations of fundamental elements. For instance, in self-driving tasks, the surrounding environment consists of elements like bicycles, pedestrians, and cars. Due to the exponential growth of possible element combinations, it is impractical to encounter and learn from every possible configuration during training. Rather than learning how to act in each unique combination, humans instead learn to interact with individual elements – such as following a car or avoiding pedestrians – and then extrapolate this knowledge to unseen combinations of elements. Therefore, it is important to study the *generalization to unseen combinations of known elements*, hereafter referred to as the out-of-combination (OOC) generalization, and to develop algorithms that can effectively handle these unseen scenarios.

Despite the success of reinforcement learning (RL) in decision-making tasks, many existing RL algorithms, particularly in offline settings, struggle to perform adeptly under state distribution shifts between training and testing, which typically occur when the learned policy visits states that differ from the data collection policy at test time (Levine et al., 2020; Kakade & Langford, 2002; Lyu et al., 2022; Schulman, 2015). While there have been works studying this problem, most of them either (1) focus on distribution shifts where the training and testing sets share the same support but different probability densities, without accounting for the presence of entirely new and unseen element combinations (Finn et al., 2017; Ghosh et al., 2022), or (2) allow unseen elements in test combinations, which makes the problem ill-posed without introducing other potentially unrealistic assumptions (Song et al., 2024; Zhao et al., 2022). As a result, these works have failed to recognize and address the critical challenge of generalization to unseen combinations of seen elements and therefore fail to capture and compose existing knowledge for these fundamental elements.

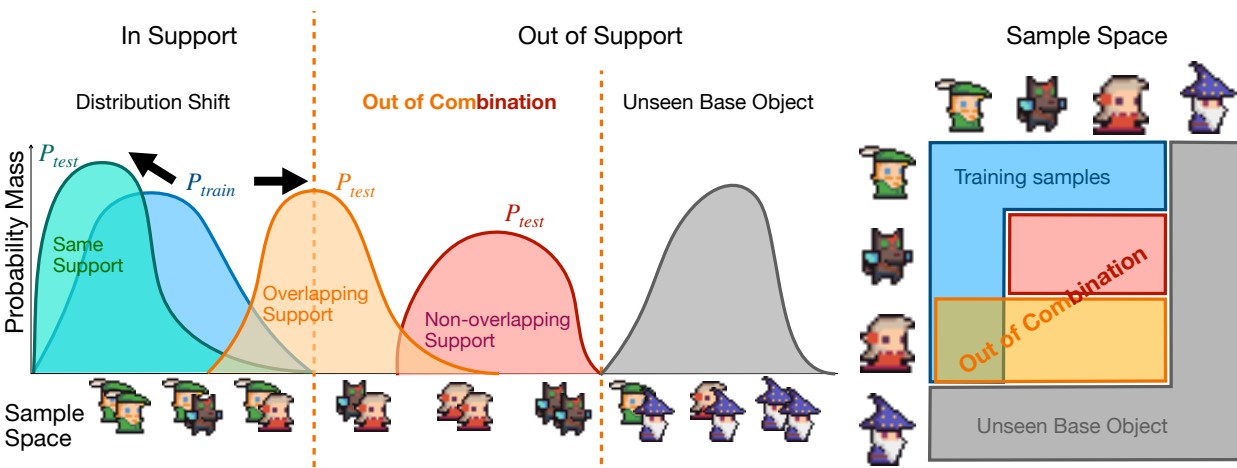

Figure 1: Different forms of out-of-distribution states. 🎃🦇🧝 are seen base elements and 🧙 is unseen base element. Their combination forms the sample space. Classic distribution shift assumes states to have the same support but different probability density. We study generalization for out-of-combination states in this work, where test time state distribution has different and possibly non-overlapping support compared to training states.

In this work, we directly tackle the problem of state combinatorial generalization in decision-making tasks, where testing states consist of unseen combinations of elements encountered during training. As illustrated in Figure 1, our task differs conceptually from traditional distribution shift problems. Unlike simple distribution shifts, where the testing set remains within the support of the training set, our proposed task requires algorithms to handle out-of-support states that are never seen during training. This makes our problem both more challenging and more representative of real-world scenarios. At the same time, our OOC setting is better defined than the unconstrained out-of-support (OOS) setting, where testing states may include completely arbitrary unseen elements and therefore is inadequately formulated and intractable without other potentially impractical assumptions such as the existence of state distance metrics (Song et al., 2024) or isomorphic Markov decision processes (MDPs) (Zhao et al., 2022). By focusing on new combinations of known elements, our setting strikes a balance between real-world applicability and tractability, making it more suitable for standardized evaluation and formal analysis.

To facilitate this study, we first provide formal definitions of state combination and OOC generalization. We then demonstrate the challenge of this task by showing how traditional RL algorithms struggle to generalize in this setting due to unreliable value prediction, and the need for a more expressive policy. On the hunt for a suitable solution, we draw inspiration from the linear manifold hypothesis in diffusion models (Chung et al., 2023; He et al., 2024b) and recent advances in combinatorial image generation (Okawa et al., 2024), and present conditional diffusion models as a promising direction by showing how they can naturally account for the combinatorial structure of states into the diffusion process, enabling better generalization in OOC settings.

Experimentally, we evaluate the models on three distinct different RL environments: maze, driving, and multiagent games. All three settings are easily adaptable to the OOC generalization problem using existing RL frameworks, demonstrating the broad applicability of the combinatorial state setup. We demonstrate behavior cloning (BC) with a conditioned diffusion model outperforms not only vanilla BC and offline RL methods like CQL (Kumar et al., 2020) but also online RL methods like PPO (Schulman et al., 2017) in zero-shot OOC generalization. To explore factors contributing to its generalization, we visualize the states predicted by the conditioned diffusion model. Our results demonstrate that the model effectively captures the core attributes of each base element and accurately composes future states by integrating these fundamental attributes. We demonstrate that, while exploration is commonly used to enhance model generalization, OOC generalization relies instead on the use of a more expressive policy.

## 2 Related Work

### 2.1 Generalization in RL

**Meta RL**   Meta RL is often seen as the problem of "learning to learn", where agents are trained on several environments sampled from a task distribution during meta-training and tested on environments sampled from the same distribution during meta-testing (Yu et al., 2020; Finn et al., 2017). In the K-shot meta-RL setting, the model can interact with the testing environment K times during meta-testing time to update the model using reward (Finn et al., 2017; Mitchell et al., 2021; Li et al., 2020; Rakelly et al., 2019). Our setting is different from Meta-RL as the training and testing environments are sampled from different distributions and conditioning is provided while restricting K to zero.

**Unsupervised RL**   Unsupervised RL aims to acquire broadly useful skills without access to task-specific rewards and then leverage these skills to accelerate or improve performance on downstream tasks with extrinsic rewards, often via an initial unsupervised "exploration" phase followed by a reward-supervised fine-tuning phase (Eysenbach et al., 2018; Laskin et al., 2022; Jaderberg et al., 2016). While our setting also assumes reuse of previously learned information and focuses on generalization, it differs in that we operate under a fully offline, reward-agnostic setup, without access to or reliance on any implicit reward signals during training.

**Zero-shot domain transfer**   The problem of zero-shot domain transfer assumes that the model is trained and tested on different domains that might have some similarities but are sampled from different underlying distributions (Kirk et al., 2023). The most widely used technique is domain randomization, approaching this problem by producing a wide range of contexts in simulation (Kirk et al., 2023; Mehta et al., 2020). However, it commonly assumes that information about the testing environment is not accessible (Mehta et al., 2020) and focuses more on sim2real problems (Kirk et al., 2023), whereas we assume test time information is given through conditioning but restricting the training set to have narrow coverage.

**Subtask and Hierarchical RL**   These two settings focus on learning reusable skills that can be sequenced to complete long horizon tasks (Parr & Russell, 1997; Lin et al., 2022; Dietterich, 2000; Nachum et al., 2018; Jothimurugan et al., 2023; Bakirtzis et al., 2024). The concept of compositionally is also a key component in subtask learning, where different sub-trajectories or intermediate goals are composed together to better perform a long horizon task (Jothimurugan et al., 2023; Lin et al., 2022; Bakirtzis et al., 2024; Mendez et al., 2022). We would like to note the difference between compositionally in trajectory stitching and our definition of state composition, where *subtasks in trajectory stitching are often data supported* as shown in Figure 2.

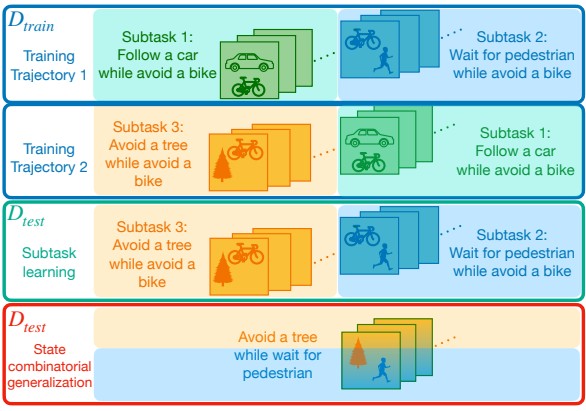

Figure 2: Visualization of states in trajectories for training, subtask learning, and state combinatorial generalization. Subtask learning involves stitching together subtask 3 in the training trajectory 2 with subtask 2 in trajectory 1. Combinatorial generalization involves simultaneously avoiding a tree and waiting for a pedestrian. Each of those two elements appeared in the training states but had never been combined.

### 2.2 Combinatorial Generalization

**In Computer Vision**   The closest line of work to ours is combinatorial generalization for image generation where the model needs to learn new combinations of a discrete set of basic concepts like color and shapes and generalize to unseen combinations (Wiedemer et al., 2024; Okawa et al., 2024; Schott et al., 2021; Hwang et al., 2023). This problem is often approached with disentangled representation learning (Liu et al., 2023; Schott et al., 2021) with models like VAE but little evidence shows they can fully exhibit generalization ability (Schott et al., 2021; Montero et al., 2020). Okawa et al. (2024) studied the capabilities of conditioned

diffusion models on a synthetic shape generation task and showed that their composition ability emerges with enough training, first to closer concepts, then to farther ones. However, we would like to emphasize the difference between image generation and decision-making tasks, where decision-making in the classic RL setting includes evaluating candidate actions, interacting with unpredictable environments, and handling future state distribution shifts caused by previous action choices and environment (Levine et al., 2020), different from image generation where it is one step and does not interact with the environment.

**In RL** Song et al. (2024) addresses the problem of generalization to unsupported states by decomposing it into the closest state in the training set and their difference, which requires the existence of a distance function to map the unseen state back to data supported region to ensure conservatism. However, we do not assume there exists a distance function between states and we do not explicitly encourage the model to be conservative. Zhao et al. (2022) uses an object oriented environment to study compositional generalization by learning the world model under the assumption that different combinations have isomorphic MDPs and objects are replaceable with each other. However, we do not assume our MDPs to be isomorphic, as each object in our setup possesses unique attributes that are non-transferable, leading to the emergence of complex underlying modalities. To the best of our knowledge, we are the first to investigate the problem of generalization to unsupported states with novel combinations of basic elements, without relying on mapping unseen states back to data-supported regions.

## 2.3 Diffusion Model for Decision making

Diffusion models emerged as a popular architecture for decision-making tasks and demonstrated superior performance compared to traditional RL algorithms, especially on long-horizon planning tasks (Janner et al., 2022; Wang et al., 2022; Liang et al., 2023a; Mishra et al., 2023). Some following work further studied conditioned diffusion models (Chi et al., 2023; Ajay et al., 2022; Li et al., 2023) and demonstrated their ability to stitch trajectories with different skills or constraints together. Application in multi-task environment (He et al., 2024a; Liang et al., 2023b) and meta-learning setting (Ni et al., 2023; Zhang et al., 2024) further demonstrate their ability to capture multi-modality information in the offline dataset.

# 3 Problem Formulation

In this section, we formally define the problem of state combinatorial generalization by providing definitions for state combination and identify out-of-combination generalization as a problem for generalization to different supports in the same sample space.

## 3.1 States Formed by Element Combinations

Following Wiedemer et al. (2024), we first denote $e \in \mathbf{E}$ to be a *base element* for an environment. A base element is defined to be the most elementary and identifiable element that is relevant to the decision making task of interest. For example, in a traffic environment, the set $\mathbf{E}$ can be the set of vehicles that can occur in the environment such as {car, bike}; and in a 2D maze environment, the set $\mathbf{E}$ can be the set of possible locations labeled by the $x, y$-axis coordinate of the agent, i.e. $\mathbb{R}^2$. Suppose there are $n$ base elements in an environment. Since these elements are the fundamental components relevant to the decision making task, we can form a *latent vector* $\boldsymbol{z} = (z_1, z_2, ..., z_n) \in \mathbf{Z} \equiv \mathbf{E}^n$, where $z_i \in \mathbf{E} \; \forall i \in \{1, \ldots, n\}$ that represents the combination of all rudimentary components appearing in this environment related to the decision making task.

Each element can also be associated with a collection of *attributes* $\boldsymbol{r}$ such as the color of the vehicle and the velocity of the agent. Attributes are components that are necessary for rendering the states but not essential for the decision making task. The *rendering function* $f(\boldsymbol{z}, (\boldsymbol{r}_1, \boldsymbol{r}_2, \ldots, \boldsymbol{r}_n))$ then maps the latent and the attributes to a state $\boldsymbol{s} \in \mathbf{S}$. In the traffic environment example, $f$ is equivalent to reconstructing the cars and the bikes given their colors and positions, etc. All reconstructed base elements collectively determine a state $\boldsymbol{s}$. Concretely, we provide the following definition:

**Definition 3.1** (States and latent vectors)**.** *For any state $\boldsymbol{s}$ with $n$ base elements in state space $\boldsymbol{S}$ and rendering function $f$, we have $\boldsymbol{s} = f(\boldsymbol{z}, (\boldsymbol{r}_1, \boldsymbol{r}_2, \ldots, \boldsymbol{r}_n))$ where the corresponding latent vector $\boldsymbol{z}$ in latent space $\boldsymbol{Z} \equiv \mathbf{E}^n$ for $\boldsymbol{s}$ is $\boldsymbol{z} = (z_1, z_2, ..., z_n)$ where $z_i \in \mathbf{E}$ for $i = 1, ..., n$.*

With our definition of base elements and states, *the combinatorial property of states naturally follows as the composition of different base elements in the latent space.*

Notice that in practice, for the same environment, one can define different base element sets depending on the desired granularity of the task. In addition, since we usually can only obtain observations of the states, in practice we can only extract the empirical latent vector $\tilde{\boldsymbol{z}}$ from the observation.

### 3.2 Generalization on Probability Space Support

Since we have identified the fundamental elements of the state in the target decision making task, we can formulate the distribution of states with the probability spaces of latent vectors. When our base element set is discrete, finite, and countable, the probability mass function (PMF) $p$ can directly ascribe a probability to a sample in $\mathbf{Z}$. Then we can define the corresponding probability space as

**Definition 3.2** (Probability space for discrete latents)**.** *Define the sample space $\mathbf{Z}$ as the set of all possible $\boldsymbol{z}$. $\sigma$-algebra $\Sigma = 2^{\mathbf{Z}}$ is the power set of $\mathbf{Z}$. $p : \mathbf{Z} \to [0, 1]$ such that $\sum_{\boldsymbol{z} \in \mathbf{Z}} p(\boldsymbol{z}) = 1$ is the PMF. Then the probability space over the latent vector $\mathbf{z}$ can be defined as $P = (\mathbf{Z}, \Sigma, p)$.*

When $\mathbf{Z}$ is a continuous space, we can also have the correspnding definitions.

**Definition 3.3** (Probability space for continuous latents)**.** *Define the sample space $\mathbf{Z}$ as the set of all possible $\boldsymbol{z}$. $\sigma$-algebra $\Sigma = \mathcal{B}(\mathbf{Z})$ is the Borel set of $\mathbf{Z}$. $p : \mathbf{Z} \to [0, 1]$ such that $\int_{\boldsymbol{z} \in \mathbf{Z}} p(\boldsymbol{z}) d\boldsymbol{z} = 1$ is the probability dense function (PDF). Then the probability space over the latent vector $\mathbf{z}$ can be defined as $P = (\mathbf{Z}, \Sigma, p)$.*

The support of $P = (\mathbf{Z}, \Sigma, p)$ can then be defined as $\operatorname{supp} P := \{\boldsymbol{z} \in \mathbf{Z} : p(\boldsymbol{z}) > 0\}$.

State combinatorial generalization, or OOC generalization, is then defined as generalizing to latent probability space with a different support. Denote the latent probability space of training states as $P_{train} = (\mathbf{Z}, \Sigma_{train}, p_{train})$ and testing states as $P_{test} = (\mathbf{Z}, \Sigma_{test}, p_{test})$, then combinatorial generalization assumes $\operatorname{supp}\{P_{train}\} \neq \operatorname{supp}\{P_{test}\}$. That is to say, combinatorial generalization in state space requires generalizing to a distribution of latent vectors with different, and possibly non-overlapping support (Wiedemer et al., 2024). Whereas traditional distribution shift in RL normally assumes different PMF or PDF ($p_{train} \neq p_{test}$), as shown in Figure 1.

### 3.3 Constraint for OOC Generalization

One crucial assumption made by OOC generalization is that all base elements are seen at training time. Recall the latent vector $\boldsymbol{z} = (z_1, z_2, ..., z_n)$ where $z_i \in \mathbf{E}$ for $i = 1, ..., n$ represent the appearing base elements for a given state. This indicates that the marginal distribution $p(z_i) > 0$ for all $z_i$ at training time, or equivalently the training probability space has full support over the marginals. For discrete latent spaces, this also implies that every base element that appeared in the training state space would appear at least once in one latent feature $\boldsymbol{z}$. To ensure full support of base elements, the union of marginal supports at test time should be a subset of that at training time. Finally, to test generalizability, we assume $\operatorname{supp}\{P_{train}\} \subsetneq \mathbf{Z}$, i.e. the training probability space doesn't have full support on the entire latent space.

**Constraint 3.4** (Combinatorial support)**.** *Given probability spaces $P = (\mathbf{Z}, \Sigma_P, p)$ and $Q = (\mathbf{Z}, \Sigma_Q, q)$ over latent vector $\boldsymbol{z} = (z_1, z_2, ..., z_n) \in \mathbf{Z}$ where $z_i \in \mathbf{E}$ for $i = 1, ..., n$, $P$ has full combinatorial support for $Q$ if: $\bigcup_{i=0}^{n}\{z_i \in \mathbf{E} : q(z_i) > 0\} \subseteq \bigcup_{i=0}^{n}\{z_i \in \mathbf{E} : p(z_i) > 0\}$.*

## 4 Why traditional RL fails

Most RL algorithms include estimating the expected cumulative reward of choosing a specific action given the current state (Schulman et al., 2017; Kumar et al., 2020; Haarnoja et al., 2018). We demonstrate the

estimation of value functions is problematic given unsupported states and this can not be solved by more exploration or more training data in this section.

## 4.1 RL and expected reward estimation

Most deep RL algorithms rely on learning a Q or Value function, which takes in the current state as network input and predicts the expected future reward (Schulman et al., 2017; Haarnoja et al., 2018). Since states with unseen composition are unsupported and fall within the undertrained regions of the neural network, the value prediction is highly unreliable. This affects both value-based methods that directly choose the maximum action with erroneous Q value and policy-based methods that update the actor with an erroneous value prediction. We plot the expected Q-values learned by CQL alongside the actual return-to-go in both failed and success scenarios in Roundabout environment (Leurent, 2018) (Section 7.1) when presented with OOC states in Figure 3. The grey dashed line is the expected Q-values the model predicts for in-distribution (ID) states. When comparing the Q-value predicted on in-distribution and OOC states:

One observation about inaccuracies of value prediction can be made: *Q function shows signs of memorizing, which assigns similar Q values for both ID and OOC states.*

The problem of distribution shift is even more challenging for offline RL, where suboptimal action prediction can cause future states to deviate away from the training set and lead to compounding errors (Levine et al., 2020). This problem can often be mitigated with a wider training state distribution under the assumption that test time states are sampled from a distribution with different probability density but same support. *However, since new states with different object combinations are out of support of the training environment, neither using a more exploratory policy nor collecting more training trajectories for offline RL can reach these OOC states, and thus the issue cannot be fundamentally resolved.* We need a policy with better generalization to unsupported states to achieve zero-shot generalization in this problem.

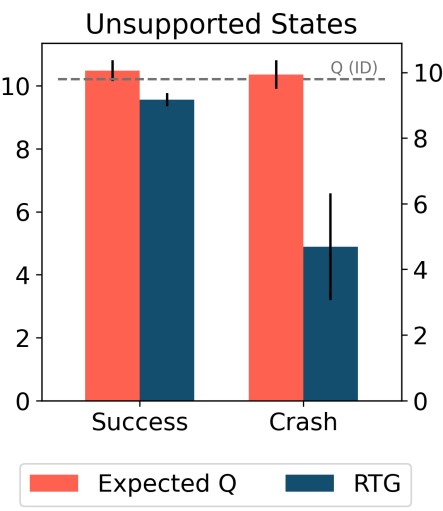

Figure 3: Expected Q value predicted by CQL and in OOC and ID states actual return-to-go (RTG) in OOC states in Roundabout environment.

# 5 Why Diffusion Models Generalize Better

In this section, we first introduce diffusion model notations and then provide a proof sketch of why diffusion models can generalize to OOC states.

## 5.1 Diffusion Models

Diffusion models are among the most popular methods for density estimation. Ho et al. (2020) proposed DDPM to model the data generation process with a forward and reverse process. In the forward process, noise is added to corrupt data $\boldsymbol{x}_t$ iteratively for $T$ timesteps towards a standard Gaussian distribution. The target of diffusion modeling is to learn the reverse process $p_\theta(\boldsymbol{x}_{t-1}|\boldsymbol{x}_t) := \mathcal{N}(\boldsymbol{x}_{t-1}; \boldsymbol{\mu}_{\boldsymbol{\theta}}(\boldsymbol{x}_t, t), \boldsymbol{\Sigma}_{\boldsymbol{\theta}}(\boldsymbol{x}_t, t))$. This way, we can sample from the data distribution by first obtaining a Gaussian noise $\boldsymbol{x}_t$ and then iteratively sampling from $p_\theta(\boldsymbol{x}_{t-1}|\boldsymbol{x}_t)$. With reparametrization trick, we can train a model $\boldsymbol{\epsilon}_{\boldsymbol{\theta}}$ to predict the noise $\boldsymbol{\epsilon}$ at each timestep $t$, and gradually denoise using update rule $\boldsymbol{x}_{t-1} = \frac{1}{\sqrt{\alpha_t}}\left(\boldsymbol{x}_t - \frac{1-\alpha_t}{\sqrt{1-\bar{\alpha}_t-\sigma_t^2}}\epsilon_\theta(\boldsymbol{x}_t, t)\right) + \sigma_t\boldsymbol{\epsilon_t}$, where $\boldsymbol{\epsilon_t} \sim \mathcal{N}(0, I)$ with variance schedulers $\alpha_t, \bar{\alpha}_t$. Given the same pretained diffusion model, one can also perform DDIM sampling (Song et al., 2020a) $\boldsymbol{x}_{t-1} = \sqrt{\alpha_{t-1}}\left(\frac{\boldsymbol{x}_t - \sqrt{1-\alpha_t}\boldsymbol{\epsilon_\theta}(\boldsymbol{x}_t, t)}{\sqrt{\alpha_t}}\right) + \sqrt{1-\alpha_{t-1}}\epsilon_\theta(\boldsymbol{x}_t, t) + \sigma_t\boldsymbol{\epsilon_t}$ to enable fast sampling.

Song et al. (2020b) formally established the connection between diffusion models and score-based stochastic differential equations (SDE). Interestingly, they discovered that each diffusion process has a corresponding probability flow ODE that shares the same intermediate marginal distributions $p(\boldsymbol{x}_t, t)$ for all $t$. The transformation between probability flow ODE and SDE can be easily achieved by adjusting the random noise hyperparameter $\sigma$ in DDIM sampling.

## 5.2 OOC Generalization in Diffusion Models

We now give a theoretical justification for OOC generalization by showing that a well-trained diffusion model concentrates probability mass on in-combination states (seen and OOC), with a quantifiable density lower bound.

Since states are formed by *combinations of base elements* (Definition 3.1), with a well-constructed and encoded **Z**, we assume that the set of in-combination states (both supported and OOC states) concentrates on a lower-dimensional manifold $\mathcal{M}$ embedded in the high dimensional ambient state space. As a result, we first adopt a **linear manifold assumption (Assumption B.1)**, where $\mathcal{M}$ is the linear manifold that the in-combination states lie along.

To connect this formulation to the diffusion sampling process, we assume we have access to a **well-trained diffusion model (Assumption B.2)** that: (i) contract the components that are orthogonal to the manifold and pull the samples towards the manifold (block-wise bi-Lipschitz with contraction), and (ii) the learned reserve mean should not mix the manifold orthogonal and on-manifold components (block preserving denoising). With above assumptions, we present the following theorem that quantifies the lower bound of the probability density assigned to a state:

**Theorem 5.1.** *Under the linear manifold assumption (Assumption B.1), and let $p_\theta$ be a DDPM diffusion using parametrization from Equation 2 that is well trained (Assumption B.2), with some constant C, for all $\boldsymbol{s} \in S$*

$$p_\theta(\boldsymbol{s}) \geq C \exp\left(-\frac{\|W_{\mathcal{M}}(\boldsymbol{s} - \mu_0)\|^2}{2(\sigma_0^{\mathcal{M}})^2} - \frac{\|W_\perp \boldsymbol{s}\|^2}{2(\sigma_0^\perp)^2}\right) \tag{1}$$

*where $0 < A_t^{\mathcal{M}} \leq B_t^{\mathcal{M}} < \infty$, $0 < A_t^\perp \leq B_t^\perp < 1$ are the block-wise bi-Lipschitz constants for the on-manifold component and the orthogonal component respectively, $W_{\mathcal{M}}$ and $W_\perp$ are projection operators onto the manifold and its orthogonal complement respectively, $\sigma_0^{\mathcal{M}}$ and $\sigma_0^\perp$ are the effective variances along and orthogonal to the manifold, $\mu_0$ is the learned noise-free center, and $C = (2\pi)^{-d/2}\left(\prod_{t=1}^{T}(B_t^{\mathcal{M}}/A_t^{\mathcal{M}})^{-m}(B_t^\perp/A_t^\perp)^{m-d}\right)(\sigma_0^{\mathcal{M}})^{-m}(\sigma_0^\perp)^{m-d}$.*

**Proof sketch:** The proof proceeds by induction over diffusion timesteps $t = T, \ldots, 0$. We begin with the base case at $t = T$ where $\mathbf{s}_T \sim \mathcal{N}(0, I)$, and establish the initial Gaussian density lower bound that factorizes along the manifold and its orthogonal complement via the projectors $(W_{\mathcal{M}}, W_\perp)$. For the inductive step from timestep $k$ to $k-1$, we decompose the DDPM reverse process $\mathbf{s}_{k-1} = \boldsymbol{\mu}_\theta(\mathbf{s}_k, k) + \sigma_k \epsilon$ into two operations: (1) applying the learned denoising mean $\boldsymbol{\mu}_\theta(\cdot, k)$, and (2) adding Gaussian noise $\sigma_k \epsilon$. For operation (1), we prove with Lemma B.3, that the bi-Lipschitz property ensures that applying $\boldsymbol{\mu}_\theta$ transforms the density bound with updated means and scales the variances. Moreover, the block-preserving property ensures that on-manifold and orthogonal components do not mix, maintaining the factorization. For operation (2), we can observe that adding the aforementioned Gaussian noise simply inflates along and orthogonal manifold components' variances by $\sigma_k^2$. Combining (1) and (2) closes the induction and gives, at $t = 0$, a block Gaussian lower bound centered at $\mu_0$ (Equation 1). We provide the full statement and the formal proof in Appendix B.

Intuitively, this theorem suggests that, the lower bound of the density assigned at a certain point by a well-trained diffusion model depends on how far it is along the manifold from the model's noise-free center and how far it is off the manifold. Because of the contraction towards the manifold, off-manifold components incurs harsh penalty under the this well-train model, with likelihood dropping rapidly as the off-manifold component grows.

# 6 Conditioned Planning with Diffusion

The theoretical analysis in the previous section provides promising signals for OOC generalization in diffusion models. However, it also suggests that reliable generalization requires the linear manifold assumption and a well-trained diffusion model that can capture the characteristics of the manifold. While in environments like a 2D maze, the linear manifold assumption can be naturally satisfied (as 2D planes through the origin are linear subspaces), for more complex environments, this assumption does not necessarily hold true without a powerful encoder that maps the element set to a proper embedding space. Moreover, the block-wise bi-Lipchitz with contraction and block preserving denoising assumption indicates that the diffusion model needs to be manifold (i.e., element combination) aware, which is also not guaranteed with vanilla unconditional diffusion models. As a result, careful model architectural designs are required to ensure empirical performance.

We take inspiration from recent computer vision research, which provides evidence of the combinatorial generalization capabilities of conditioned diffusion models in more complicated data spaces: Aithal et al. (2024) identifies the phenomena where diffusion models generate samples out of the support of training distribution through interpolating different complex modes on a data manifold. Kadkhodaie et al. (2023) demonstrate generalization to unsupported data by showing two diffusion models trained on large enough non-overlapping data converge to the same denoising function. Okawa et al. (2024) showed that given different concepts like shape, color, and size in synthetic shape generation, conditional diffusion models demonstrate a multiplicative emergence of combinatorial abilities where it will first learn how to generalize to concepts closer to the training samples (i.e. only change one of color, shape, and size) and eventually adopt full compositional generalization ability with enough training.

Therefore, we propose to use a conditioned diffusion model with learnable element embeddings to tackle the OOC state problem for decision-making tasks. Following the trajectory-based planning formulation of Janner et al. (2022), we train a conditional diffusion model to denoise (predict) future state-action pairs conditioned on the current observation. Importantly, this training is carried out without reward guidance; instead, it is trained via behavior cloning on successful demonstrations collected by a PPO (Schulman et al., 2017) model in the in-distribution environment, capturing the relationship between states, successful actions, and future states. Conditioning is applied in two complementary ways. First, we enforce consistency of the generated trajectory with the current observation by replacing the corresponding component of the predicted trajectory with the actual environment observation at every denoising step. Second, to enhance this modeling, we incorporate a cross-attention layer after each residual block in the Diffusion UNet, allowing the predicted trajectory to directly attend to the latent vector $z$ contained in the observation of the current timestep. Consequently, when conditioned on novel combinations of base elements, the generated trajectories remain dynamically consistent with the underlying dynamics (MDP), facilitating robust generalization to novel scenarios. For deployment, the first action in the generated trajectory is then used to step the environment (Appendix C.4). Detailed architecture of the model can be found in Appendix C.3 and the planning process is described in Algorithm 1.

Note that in practice, our conditioned diffusion model and the element embeddings are trained end-to-end, and thus the resulting learned model does not necessarily fit the theoretical assumptions above. However, as we will show in the next section, empirically this model demonstrates successful OOC generalization in various RL environments.

# 7 Experiments

The primary goal of our experiments is to answer the following questions: (1) (Wide applicability) Does the state-space of different existing RL environments exhibit a compositional nature? (2) (Advantages) What are some interesting features conditional diffusion models have that contribute to their performance when generalizing to OOC states? (3) (Conditioning) Does conditioning help with OOC generalization?

---

**Algorithm 1** Planning with Attention-based Composition Conditioned Diffusion Model

---

**Input:** Diffusion model $\epsilon_{\boldsymbol{\theta}}$, compositional elements extractor $r$, learnable embedding function $h$, classifier-free guidance scale $\lambda$, state dimentionality $d_S$, initial observation $\boldsymbol{o}$, environment simulator $env$

**while** not done **do**

    Initialize $\boldsymbol{s}_t \sim \mathcal{N}(0, I)$

    $\boldsymbol{c} \leftarrow r(\boldsymbol{o})$                                                      ▷ Extract observed compositional information

    $\boldsymbol{z} \leftarrow h(\boldsymbol{c})$                                                           ▷ Obtain element embedding

    **for** $t \leftarrow T, ... 1$ **do**

        $\boldsymbol{s}_t[: d_S, 0] \leftarrow \boldsymbol{o}$                                   ▷ (Cond state) Replace the first denoised state with observed $\boldsymbol{o}$

        $\widetilde{\boldsymbol{\epsilon}_t} = (1 + \lambda)\epsilon_{\boldsymbol{\theta}}(\boldsymbol{s}_t, \boldsymbol{z}, t) - \lambda \epsilon_{\boldsymbol{\theta}}(\boldsymbol{s}_t, t)$             ▷ (Cond z) Classifier free guidance

        $\boldsymbol{x}_{t-1} = \frac{1}{\sqrt{\alpha_t}}\left(\boldsymbol{s}_t - \frac{1-\alpha_t}{\sqrt{1-\bar{\alpha}_t}}\widetilde{\boldsymbol{\epsilon}_t}\right) + \sigma_t \boldsymbol{\epsilon_t}, \text{ where } \boldsymbol{\epsilon_t} \sim \mathcal{N}(0, I)$

    **end for**

    $\boldsymbol{a} \leftarrow \boldsymbol{x_0}[d_S :, 0]$                                                              ▷ Extract action

    $\boldsymbol{o} \leftarrow env.step(a)$

**end while**

---

## 7.1 Single-agent Environment

**Environment** HighwayEnv (Leurent, 2018) is a self-driving environment where the agent needs to control a vehicle to navigate between traffic controlled by predefined rules. We specifically look at the Roundabout environment with two types of traffic: cars and bicycles (Visualization in Appendix C.6.1).

State in this environment is a composition of four environment vehicles that are either cars or bicycles and the ego agent, which is always a car. Environment observation contains observability, the locations and speed of the ego and surrounding agent, and *whether this agent is a car or a bike (Conditioning)*. During training time, the environment will only generate traffic of all cars or all bicycles with equal probability. During test time, environments will generate a mixture of cars and bicycles (detailed setup in Appendix C.7). Cars and bicycles have different sizes, max speeds, and accelerations, leading to different behavior patterns. This is an instance of generalizing to OOC states with non-overlapping support.

**Results** As shown in Figure 4, the conditional diffusion model has almost half the number of crashes and higher reward when zero-shot generalizing to states with mixture traffic. Results of models with different sizes are shown in Table 5 to eliminate the concern of performance gain due to larger model sizes. Since we train the diffusion model exclusively on successful PPO trajectories, the training state distribution for diffusion is much narrower compared to that of other online methods. This is particularly interesting since it is widely acknowledged that online models have better generalization compared to offline models (Levine et al., 2020).

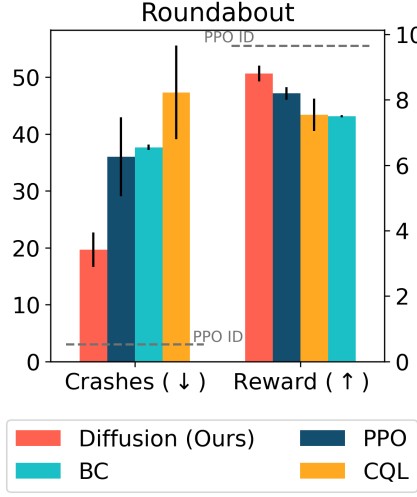

Figure 4: Total number of crashes and average reward for BC(MLP), PPO, CQL, and diffusion model in the testing environment.

> **Takeaway 1**: Conditional diffusion models, trained on an offline dataset with narrow state distribution with full combinatorial generalization support, have better zero-shot generalization performance to OOC states compared to online RL trained in the same environment.

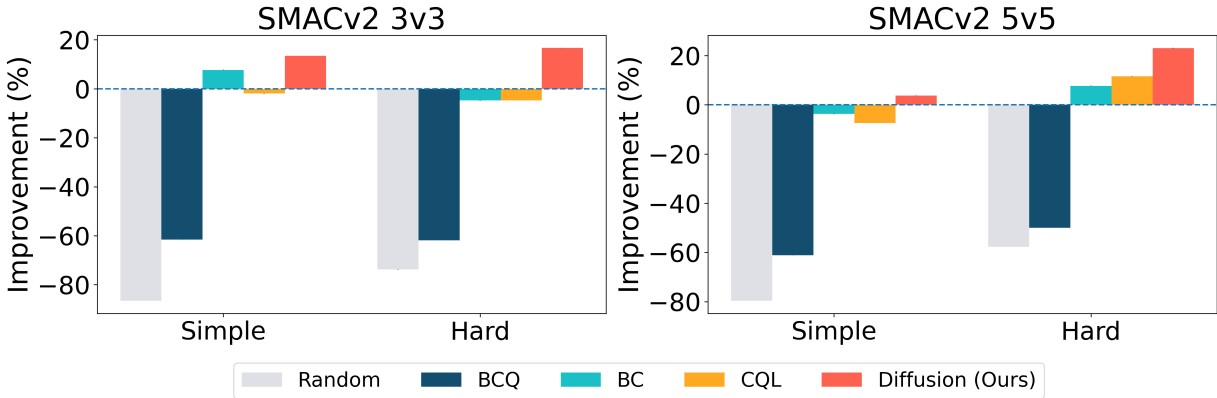

Figure 5: Relative improvement % compared to MAPPO on two SMACv2 scenarios: 3v3 and 5v5. Conditional Diffusion shows large and consistent improvements over the MAPPO and other BC and RL baselines, especially in the hard scenario, where we train on teams with the same unit type only but test on random team compositions.

## 7.2 Multi-agent Environment

**Environment**   The StarCraft Multi-Agent Challenge (SMAC/SMACv2) (Samvelyan et al., 2019; Ellis et al., 2022) is a multi-agent collaborative game that takes several learning agents, each controlling a single army unit, to defeat the enemy units controlled by the built-in heuristic AI. This benchmark is particularly challenging for its diverse army unit behaviors and complex team combinations, which enable diverse strategies like focus fire and kiting enemies to emerge (Ellis et al., 2024). Each agent's observation includes health, shield, position, *unit type (Conditioning)* of its own, visible teammates, and enemies.

We treat one agent as the ego agent, and consider its teammates and enemies as part of the environment. Then states can be naturally seen as compositions of the unit types in a particular playthrough. *We expect the ego agent to generate different policies when playing with or against different types of units, and we aim to test OOC generalization by changing the unit composition in the environment.* Since we use a MAPPO (Yu et al., 2022) for data collection, we report the performance gain/loss compared to MAPPO as shown in Figure 5. To treat the teammates and enemies of one particular agent as environment and change their combination, we control one unit with a conditional diffusion model and let MAPPO control the rest of its teammates.

**Setup**   The unit types in this experiment are Protoss.Stalker, Protoss.Zealot, and Protoss.Colossus, referred to as $a, b, c$ respectively. We evaluate on two OOC scenarios: (1) *(Simple: Different but overlapping support)*: Train the model on randomly generated combinations ($ABC$) of all units and test it where all the units on the team have same type ($AAA$), (2) *(Hard: Non-overlapping support)*: the opposite scenario, where we train on teams with only one unit type ($AAA$), but during test-time we see any composition of these three units ($ABC$). More information about our setup could be found in Appendix C.8.

**Results**   As shown in Table 9 and Table 10, MAPPO performance drastically dropped in the hard OOC scenario by 55.2% for 5v5 and 33.3% for 3v3. If we substitute one agent generated by MAPPO with conditional diffusion, the success rate can be improved by 16.7% for 3v3 and 23.1% for 5v5 in hard OOC scenario as shown in Figure 5.

> **Takeaway 2**: Multi-agent RL, viewed from the perspective of a single ego agent, naturally requires combinatorial generalization to collaborate/compete with different agent types.

## 7.3 How Do Conditional Diffusion Models Generalize to OOC States?

To see how diffusion models generalize to OOC states, we render the states predicted by the diffusion models given different conditionings with the same current state, as shown below in Figure 6.

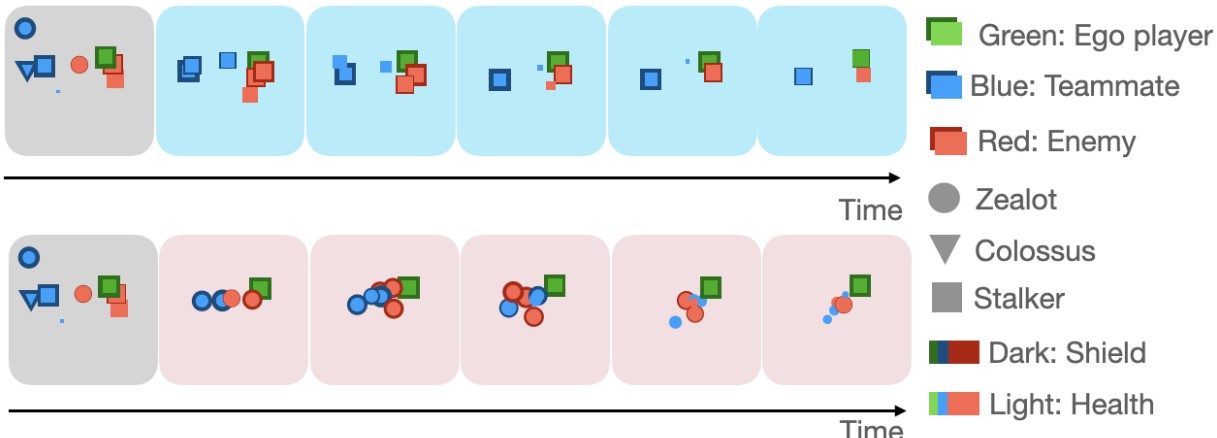

Figure 6: Rendering of future states predicted by the diffusion model given different conditionings. The grey box is the current state. Blue backgrounds are conditional on all Squares (long attack range) and pink backgrounds are conditioned on all circles (short attack range). Smaller sizes represent less shield and health. More examples shown in Appendix C.8.6.

We can see that *player type conditionings determine the unit type of agent predicted by the diffusion model and also their behavior pattern. Whereas the current state determines other attributes like initial location and health.* Different player type conditionings will lead to different strategies. The circle unit has attack range 1 and the square unit has attack range 6. For units with short attack ranges, the optimal strategy is to approach their enemies before initiating an attack. Conversely, agents with large attack ranges are advised to attack their enemies from a distance to ensure their own safety. Figure 6 shows that if we condition on all circles, the diffusion model thinks players will form a cluster and if condition on all squares, it will predict the players to attack each other from a distance, aligning well with the optimal policy. This demonstrates conditioned diffusion models' ability to *implicitly decompose states to learn underlying compositions* and *capture multimodality of different unit behavior* in the training data. It also demonstrates its ability to perform state stitching to accurately predict the underlying MDP for an unseen combination.

> **Takeaway 3**: Conditioned diffusion models show significant promise by effectively decomposing and capturing modes of individual base elements and performing state stitching, which helps them to generalize to OOC scenarios.

## 8 Ablations

In this section, we ablate over our design choices to (1) show the necessity of using the inductive bias of the latent vector as conditioning, (2) different model architectures to incorporate conditioning information

### 8.1 Necessity of Combinatorial Inductive Bias

We compare trajectories generated by the conditioned and unconditioned diffusion models in this section to demonstrate the importance of using the latent vector $z$ that contains the combinatorial latent information as conditioning. In Maze2D (Fu et al., 2020), we formulate the navigation problem as a one-step generation process where the diffusion model learns how to generate an entire valid trajectory without rolling out the current action and replanning. Since there is only one planning step in this process, the generated trajectory can be seen as the "state" in this setting, where unseen trajectories correspond to unseen states instead of time-horizon trajectory stitching. The inductive bias we use is every training trajectory will pass through three waypoints that equally slice the trajectory. In this case, the set of all waypoints forms the base element set and their combination is the latent vector that determines the shape of a generated maze trajectory. During training, we extract three points that equally slice the trajectory and use them as conditioning. During test time, we specify a new combination of three waypoints we want the generated trajectory to pass.

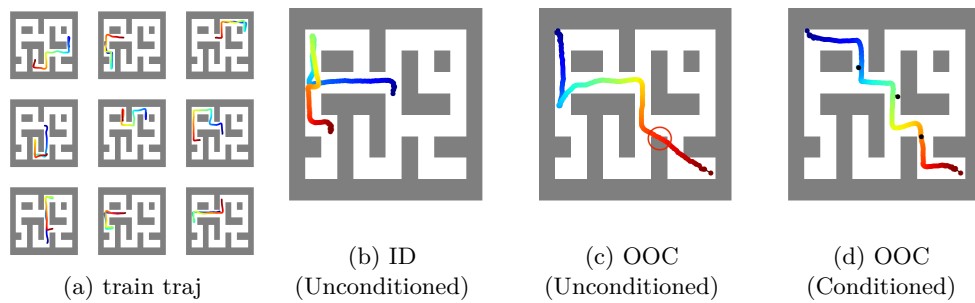

(a) train traj     (b) ID (Unconditioned)     (c) OOC (Unconditioned)     (d) OOC (Conditioned)

Figure 7: Trajectories generated in Maze2D for large maze. (a) Samples from the training set. (b) Trajectories generated by the unconditioned diffusion model given in distribution start and end positions. (c) Trajectories generated by the unconditioned diffusion model on unseen start and end positions. (d) Trajectories generated by a conditioned diffusion model using 3 waypoints (black dots) as conditioning. For results in the medium maze please refer to Appendix C.5.1.

We see that the unconditioned diffusion model successfully generated a trajectory if the start and end positions are in the training set (Figure 7b), but failed to do so given unseen start and end positions (Figure 7c). We compare the trajectories generated by the unconditioned and conditioned diffusion model given unseen start and end positions in Figure 7c and 7d. The trajectory generated by the conditioned diffusion model accurately follows the given OOC waypoints and is drastically different compared to the one without conditioning. This highlights the conditioned diffusion model's ability to generalize to OOC conditioning and its strong accuracy in adhering to the provided conditioning.

## 8.2 Model Architecture: Attention vs Concatenation

We also ablate over different model architectures: (1) concatenating the latent vector $\mathbf{z}$ with diffusion's time embedding, (2) performing cross attention between $\mathbf{z}$ and output of each Unet residual block (Architecture in Figure 9). Figure 8 shows our result: in general conditioned diffusion models outperform unconditioned ones and attention outperforms concatenation in 3 out of 4 cases.

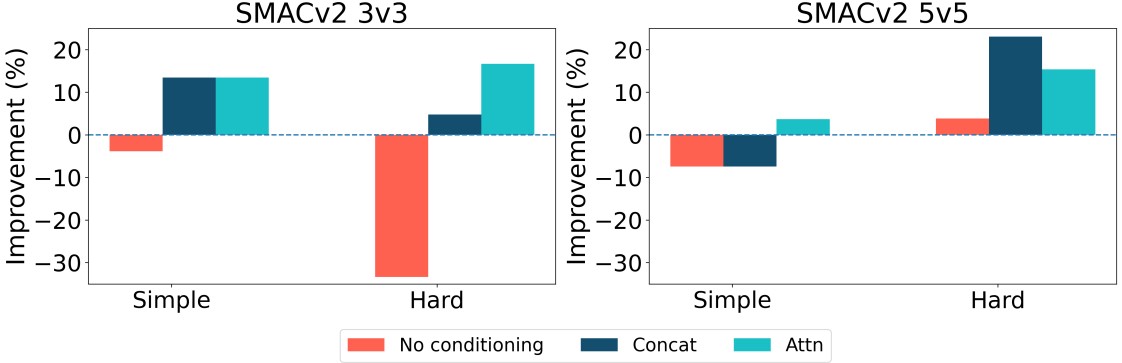

Figure 8: Improvement percentage over MAPPO for different types of conditioning in SMACv2.

**Ablation Takeaway**: Conditioned diffusion models, provided with information about the new composition of state, are able to generalize to out-of-combination conditioning and accurately adhere to the provided conditioning. Also, cross-attention with the condition vector outperforms simply concatenating it with the time-embedding in most cases.

## 9 Conclusions

Despite the success of traditional RL models in decision-making tasks, they still struggle to generalize to unseen state inputs. Most existing work focuses on RL generalization under the assumption that generalization to a different probability density function with the same support. However, we take it further and

study the problem of generalization to out-of-support states, out of combination in particular, hoping the model can exploit the compositional nature of our world. We showed how this task is challenging for value-based RL and also how conditioned diffusion models can generalize to unsupported samples. We compare the models in different environments with detailed ablation and analysis, demonstrating how each of these classic environments can be formulated as a state combinatorial problem.

However, one limitation of our setup is we model combinatorial generalization in state space as a combination of base elements, which is valid for many real-world applications but does not cover all cases. Additionally, the model has difficulty with zero-shot generalization to unseen base objects. Another constraint is efficiency, as planning with diffusion models in stochastic environments requires denoising a trajectory at each planning step, which can be computationally intensive. Various approaches have been proposed to accelerate the sampling process in diffusion models via advanced ODE solvers or knowledge distillation techniques (Song et al., 2020a; Karras et al., 2022; Kim et al., 2023; Song et al., 2023). Incorporating these works into diffusion planning would be a promising future direction to address rollout inefficiency.

### Broader Impact Statement

Developing data-driven decision-making models carries the risk of generating inappropriate or harmful actions. This work presents a conditioned model that can be manipulated through carefully forged conditioning, potentially leading to malicious actions. Our experiments were conducted in simulated environments, and testing this model in the real world without safety restrictions could be dangerous.

### Acknowledgments

This work was supported in part by the U.S. Army Research Office and the U.S. Army Futures Command under Contract No.W519TC-23-C-0030, and also by DSTA. This work was also supported in part of the ONR grant N000142312368.

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

# A Related Work

We introduced related problem setup and generalization methods in Section 2. Attached below is a review of RL environments for generalization.

## A.1 Generalization in RL

**Environments**   Most RL environments that test model generalization can be grouped into different reward functions (Rajeswaran et al., 2017; Zhang et al., 2018; Rakelly et al., 2019; Finn et al., 2017) or transition functions (Dennis et al., 2020; Machado et al., 2018; Packer et al., 2018; Zhang et al., 2018), goals or tasks (Finn et al., 2017; Yu et al., 2020), states (Nichol et al., 2018; Cobbe et al., 2019; Juliani et al., 2019; Küttler et al., 2020; Grigsby & Qi, 2020; Hansen et al., 2021; Mees et al., 2022; Cobbe et al., 2020). For environments with different state distributions, randomization (Grigsby & Qi, 2020) and procedural generation (Nichol et al., 2018; Küttler et al., 2020; Cobbe et al., 2020) are widely used to generate new states. Some vision-based environments (Juliani et al., 2019; Hansen et al., 2021; Mees et al., 2022) also use different rendering themes or layouts to generate unseen observations, more targeting sim2real problems. For robotics benchmarks like Metaworld (Yu et al., 2020) and RLbench (James et al., 2020), how much structure is shared between tasks like open a door and open a drawer is ambiguous (Ahmed et al., 2020). Also, benchmarks like Franka Kitchen (Gupta et al., 2019) focus on composing tasks at time horizons, requiring the model to concatenate trajectories corresponding to different subtasks. However, despite the large volume of generalization benchmarks, there is no benchmark designed for state combinatorial generalization to our best knowledge.

# B Proof of Theorem 5.1

In this section, we provide the full statement and proofs of Theorem 5.1. First, we establish the following assumptions.

Practically, when the element set is discrete, an embedding $\phi : \mathbf{E}^n \to \mathbb{R}^m$ is applied to map a discrete representation of latent elements to a continuous embedding space.

**Assumption B.1** (Linear manifold assumption). *Assume the states lie along a linear manifold $\mathcal{M}$ in the ambient state space $\mathbf{S} \subset \mathbb{R}^d$ and the encoded latent space $\phi(\mathbf{Z}) \subset \mathbb{R}^m$ with $m < d$ is well constructed so that $\phi(\mathbf{Z})$ is (affine) isomorphic to $\mathcal{M}$.*

Here we can interpret the linear manifold $\mathcal{M}$ as the subspace containing all "in-combination" (i.e. in support or out-of-combination) data points. Recall we can parametrize the DDPM learned reverse process of a diffusion model as

$$p_\theta(\boldsymbol{x}_{t-1}|\boldsymbol{x}_t) := \mathcal{N}(\boldsymbol{x}_{t-1}; \boldsymbol{\mu_\theta}(\boldsymbol{x}_t, t), \sigma_t I) \tag{2}$$

where $\sigma_t$ is a pre-fixed noise schedule. Given Assumption B.1, we denote the projection operation onto the manifold $\mathcal{M}$ as $W_\mathcal{M}$ and the projector onto its orthogonal complement $\mathcal{M}^\perp$ as $W_\perp$. We then further assume the diffusion model is well-trained, in particular, it should satisfy the following assumptions:

**Assumption B.2** (Well-trained diffusion model). *We assume access to a well-trained DDPM diffusion model that satisfies:*

1. ***Block-wise bi-Lipschitz with contraction:*** *For all $\boldsymbol{x}, \boldsymbol{y} \in \mathbf{S}$ and all timesteps $t$, there exists constants*

$$0 < A_t^\mathcal{M} \leq B_t^\mathcal{M} < \infty, \qquad 0 < A_t^\perp \leq B_t^\perp < 1 \tag{3}$$

   *such that*

$$\begin{aligned} A_t^\mathcal{M} \|W_\mathcal{M}(\boldsymbol{x} - \boldsymbol{y})\| &\leq \|W_\mathcal{M}(\boldsymbol{\mu_\theta}(\boldsymbol{x}, t) - \boldsymbol{\mu_\theta}(\boldsymbol{y}, t))\| \leq B_t^\mathcal{M} \|W_\mathcal{M}(\boldsymbol{x} - \boldsymbol{y})\| \\ A_t^\perp \|W_\perp(\boldsymbol{x} - \boldsymbol{y})\| &\leq \|W_\perp(\boldsymbol{\mu_\theta}(\boldsymbol{x}, t) - \boldsymbol{\mu_\theta}(\boldsymbol{y}, t))\| \leq B_t^\perp \|W_\perp(\boldsymbol{x} - \boldsymbol{y})\| \end{aligned} \tag{4}$$

2. ***Block preserving denoising:*** *For all timesteps $t$, $\boldsymbol{\mu_\theta}(\cdot, t)$ preserves the block decomposition. I.e.*

$$W_\mathcal{M}\boldsymbol{\mu_\theta}(W_\perp \boldsymbol{x}, t) = W_\perp \boldsymbol{\mu_\theta}(W_\mathcal{M}\boldsymbol{x}, t) = \mathbf{0} \tag{5}$$

   *for all $x \in \mathbf{S}$.*

The first assumption suggests that a well-trained diffusion model will contract the components that are orthogonal to the manifold and pull the samples towards the manifold (i.e. towards "in-combination" samples). The second assumption indicates that the learned reverse mean should not mix the two orthogonal components.

Based on these two assumptions, we can now prove the following lemmas that will be helpful for our proof to Theorem 5.1.

**Lemma B.3.** *If $X \in \mathbb{R}^d$ has density lower bound*

$$p_X(\boldsymbol{x}) \geq c(2\pi)^{-d/2}(\sigma^\mathcal{M})^{-m}(\sigma^\perp)^{m-d} \exp\left(-\frac{\|W_\mathcal{M}(\boldsymbol{x} - \mu)\|^2}{2(\sigma^\mathcal{M})^2} - \frac{\|W_\perp(\boldsymbol{x} - \mu)\|^2}{2(\sigma^\perp)^2}\right) \tag{6}$$

*Let g be a function that satisfies block-wise bi-Lipschitz with contraction and block reservation from Assumption B.2 with Lipschitz constants $A^\mathcal{M}, B^\mathcal{M}, A^\perp, B^\perp$ and $Y = g(X)$, then for constant c we can have*

$$p_Y(\boldsymbol{y}) \geq c(2\pi)^{-d/2}(\sigma^\mathcal{M} B^\mathcal{M})^{-m}(\sigma^\perp B^\perp)^{m-d} \exp\left(-\frac{\|W_\mathcal{M}(\boldsymbol{y} - g(\mu))\|^2}{2(\sigma^\mathcal{M} A^\mathcal{M})^2} - \frac{\|W_\perp(\boldsymbol{y} - g(\mu))\|^2}{2(\sigma^\perp A^\perp)^2}\right) \tag{7}$$

*Proof.* Because $g$ is block-wise bi-Lipschitz, $g$ is also injective and differentiable almost everywhere by Rademacher's theorem. Then using the exact density transform, for almost everywhere $\boldsymbol{y}$, we can have $\boldsymbol{x}^* = g^{-1}(\boldsymbol{y})$ and

$$p_Y(\boldsymbol{y}) = \frac{p_X(\boldsymbol{x}^*)}{|\det Dg(\boldsymbol{x}^*)|} \tag{8}$$

Equation 8 entails that in order to obtain a lower bound for $p_Y(\boldsymbol{y})$, we should obtain a lower bound for $p_X(\boldsymbol{x}^*)$ and an upper bound for $|\det \nabla_{\boldsymbol{x}^*} g(\boldsymbol{x}^*)|$.

From Equation 6 we can directly obtain

$$p_X(\boldsymbol{x}^*) \geq c(2\pi)^{-d/2}(\sigma^{\mathcal{M}})^{-m}(\sigma^{\perp})^{m-d} \exp\left(-\frac{\|W_{\mathcal{M}}(\boldsymbol{x}^* - \mu)\|^2}{2(\sigma^{\mathcal{M}})^2} - \frac{\|W_{\perp}(\boldsymbol{x}^* - \mu)\|^2}{2(\sigma^{\perp})^2}\right) \tag{9}$$

By the Liptschtiz condition we also know that

$$A^{\mathcal{M}}\|W_{\mathcal{M}}(\boldsymbol{x}^* - \mu)\| \leq \|W_{\mathcal{M}}(g(\boldsymbol{x}^*) - g(\mu))\| \Rightarrow \|W_{\mathcal{M}}(\boldsymbol{x}^* - \mu)\| \leq \frac{\|W_{\mathcal{M}}(\boldsymbol{y} - g(\mu))\|}{A^{\mathcal{M}}}$$

$$A^{\perp}\|W_{\perp}(\boldsymbol{x}^* - \mu)\| \leq \|W_{\perp}(g(\boldsymbol{x}^*) - g(\mu))\| \Rightarrow \|W_{\perp}(\boldsymbol{x}^* - \mu)\| \leq \frac{\|W_{\perp}(\boldsymbol{y} - g(\mu))\|}{A^{\perp}} \tag{10}$$

So

$$p_X(\boldsymbol{x}^*) \geq c(2\pi)^{-d/2}(\sigma^{\mathcal{M}})^{-m}(\sigma^{\perp})^{m-d} \exp\left(-\frac{\|W_{\mathcal{M}}(\boldsymbol{y} - g(\mu))\|^2}{2(\sigma^{\mathcal{M}}A^{\mathcal{M}})^2} - \frac{\|W_{\perp}(\boldsymbol{y} - g(\mu))\|^2}{2(\sigma^{\perp}A^{\perp})^2}\right) \tag{11}$$

For $|\det Dg(\boldsymbol{x}^*)|$, by block preservation, we can have

$$Dg(\boldsymbol{x}^*) = \begin{bmatrix} W_{\mathcal{M}}Dg(\boldsymbol{x}^*)W_{\mathcal{M}} & 0 \\ 0 & W_{\perp}Dg(\boldsymbol{x}^*)W_{\perp} \end{bmatrix} \tag{12}$$

Now let $v \in \mathcal{M}$ be a unit vector and $\Delta$ be a scalar, then by Lipschtiz condition, we can have

$$\|W_{\mathcal{M}}(g(\boldsymbol{x}^* + \Delta v) - g(\boldsymbol{x}^*))\| \leq B^{\mathcal{M}}\|W_{\mathcal{M}}(\Delta v)\| = B^{\mathcal{M}}|\Delta| \tag{13}$$

With $|\Delta| \to 0$, we can have $\|W_{\mathcal{M}}Dg(\boldsymbol{x}^*)v\| \leq B^{\mathcal{M}}$. Since this is true for all unit vector $v \in \mathcal{M}$, denoting $\lambda_i^{\mathcal{M}}$ as the singular values of $W_{\mathcal{M}}Dg(\boldsymbol{x}^*)W_{\mathcal{M}}$, then for all $i \in [1, 2, \ldots, m]$,

$$B^{\mathcal{M}} \geq \sup_{\|v\|=1} \|W_{\mathcal{M}}Dg(\boldsymbol{x}^*)W_{\mathcal{M}}v\| \geq s_i^{\mathcal{M}} \tag{14}$$

Similarly, for singular values $\lambda_j^{\perp}$ of $W_{\perp}Dg(\boldsymbol{x}^*)W_{\perp}$, we can have

$$B^{\perp} \geq \sup_{\|v\|=1} \|W_{\perp}Dg(\boldsymbol{x}^*)W_{\perp}v\| \geq s_j^{\perp} \tag{15}$$

for all $j \in [1, 2, \ldots, d - m]$.

Using Equation 12,14,15, we obtain

$$|\det Dg(\boldsymbol{x}^*)| = |\det W_{\mathcal{M}}Dg(\boldsymbol{x}^*)W_{\mathcal{M}}||\det W_{\perp}Dg(\boldsymbol{x}^*)W_{\perp}| = \prod_{i=1}^{m}\lambda_i^{\mathcal{M}} \prod_{j=1}^{d-m}\lambda_j^{\perp} \leq (B^{\mathcal{M}})^m(B^{\perp})^{d-m} \tag{16}$$

Putting everything together, we can have

$$p_Y(\boldsymbol{y}) \geq c(2\pi)^{-d/2}(\sigma^{\mathcal{M}}B^{\mathcal{M}})^{-m}(\sigma^{\perp}B^{\perp})^{m-d} \exp\left(-\frac{\|W_{\mathcal{M}}(\boldsymbol{y} - g(\mu))\|^2}{2(\sigma^{\mathcal{M}}A^{\mathcal{M}})^2} - \frac{\|W_{\perp}(\boldsymbol{y} - g(\mu))\|^2}{2(\sigma^{\perp}A^{\perp})^2}\right) \tag{17}$$

$\square$

To further facilitate the proof of Theorem 5.1, we further define the recursive mean as

$$\mu_T = 0, \qquad \mu_t = \mu_\theta(\mu_{t+1}, t) \tag{18}$$

and the block variance scales as

$$\sigma_T^{\mathcal{M}} = \sigma_T^\perp = \sigma_T = 1, \qquad (\sigma_{t-1}^{\mathcal{M}})^2 = \sigma_t^2 + (A_t^{\mathcal{M}})^2(\sigma_t^{\mathcal{M}})^2, \qquad (\sigma_{t-1}^\perp)^2 = \sigma_t^2 + (A_t^\perp)^2(\sigma_t^\perp)^2 \tag{19}$$

Now we are ready to prove our Theorem 5.1. Before that, we first provide the formal statement of Theorem 5.1.

**Theorem B.4** (Formal statement of Theorem 5.1). *Under Assumption B.1, and let $p_\theta$ be a DDPM diffusion using parametrization from Equation 2 that satisfies Assumption B.2, with some constant C, for all $\boldsymbol{s} \in S$*

$$p_\theta(\boldsymbol{s}) \geq C \exp\left(-\frac{\|W_{\mathcal{M}}(\boldsymbol{s} - \mu_0)\|^2}{2(\sigma_0^{\mathcal{M}})^2} - \frac{\|W_\perp \boldsymbol{s}\|^2}{2(\sigma_0^\perp)^2}\right) \tag{20}$$

*Proof.* Denoting $p_\theta(\boldsymbol{s}_t, t)$ as the density at time $t$ for noisy sample $\boldsymbol{s}_t$, we prove the statement by induction.

1. **Base Case:** At $t = T$, $\boldsymbol{s}_T \sim \mathcal{N}(0, I)$, therefore

$$p_\theta(\boldsymbol{s}_T, T) = (2\pi)^{-d/2} \exp\left(-\frac{\|W_{\mathcal{M}}\boldsymbol{s}_T\|^2}{2} - \frac{\|W_\perp \boldsymbol{s}_T\|^2}{2}\right) \tag{21}$$

2. **Inductive Hypothesis:** Assume that for timestep $k$,

$$p_\theta(\boldsymbol{s}_k, k) \geq (2\pi)^{-d/2}\left(\prod_{t=k+1}^T (\frac{B_t^{\mathcal{M}}}{A_t^{\mathcal{M}}})^{-m}(\frac{B_t^\perp}{A_t^\perp})^{m-d}\right)(\sigma_k^{\mathcal{M}})^{-m}(\sigma_k^\perp)^{m-d}\exp\left(-\frac{\|W_{\mathcal{M}}(\boldsymbol{s} - \mu_k)\|^2}{2(\sigma_k^{\mathcal{M}})^2} - \frac{\|W_\perp(\boldsymbol{s} - \mu_k)\|^2}{2(\sigma_k^\perp)^2}\right) \tag{22}$$

3. **Induction Step:** For timestep $k - 1$, recall $\boldsymbol{s}_{k-1} = \mu_\theta(\boldsymbol{s}_k, k) + \sigma_k\epsilon$ for $\epsilon \sim \mathcal{N}(0, I)$, we can break this down into two operations: applying $\mu_\theta(\cdot, k)$ and adding Gaussian white noise.

Since $\mu_\theta$ satisfies Assumption B.2, let $\boldsymbol{y} = \mu_\theta(\boldsymbol{s}_k, k)$, we can directly apply Lemma B.3 and obtain

$$p_Y(\boldsymbol{y}) \geq c(2\pi)^{-d/2}(\sigma_k^{\mathcal{M}}B_k^{\mathcal{M}})^{-m}(\sigma_k^\perp B_k^\perp)^{m-d}\exp\left(-\frac{\|W_{\mathcal{M}}(\boldsymbol{y} - \mu_\theta(\mu_k, k))\|^2}{2(\sigma_k^{\mathcal{M}}A_k^{\mathcal{M}})^2} - \frac{\|W_\perp(\boldsymbol{y} - \mu_\theta(\mu_k, k))\|^2}{2(\sigma_k^\perp A_k^\perp)^2}\right) \tag{23}$$

where $c = \prod_{t=k+1}^T (\frac{B_t^{\mathcal{M}}}{A_t^{\mathcal{M}}})^{-m}(\frac{B_t^\perp}{A_t^\perp})^{m-d}$. Notice that this is equivalent to

$$p_Y(\boldsymbol{y}) \geq c(2\pi)^{-d/2}(\sigma_k^{\mathcal{M}}B_k^{\mathcal{M}})^{-m}(\sigma_k^\perp B_k^\perp)^{m-d}\exp\left(-\frac{\|W_{\mathcal{M}}(\boldsymbol{y} - \mu_\theta(\mu_k, k))\|^2}{2(\sigma_k^{\mathcal{M}}A_k^{\mathcal{M}})^2} - \frac{\|W_\perp(\boldsymbol{y} - \mu_\theta(\mu_k, k))\|^2}{2(\sigma_k^\perp A_k^\perp)^2}\right)$$

$$= (2\pi)^{-d/2}\prod_{t=k+1}^T (\frac{B_t^{\mathcal{M}}}{A_t^{\mathcal{M}}})^{-m}(\frac{B_t^\perp}{A_t^\perp})^{m-d}(\sigma_k^{\mathcal{M}}B_k^{\mathcal{M}})^{-m}(\sigma_k^\perp B_k^\perp)^{m-d}$$

$$\exp\left(-\frac{\|W_{\mathcal{M}}(\boldsymbol{y} - \mu_{k-1})\|^2}{2(\sigma_k^{\mathcal{M}}A_k^{\mathcal{M}})^2} - \frac{\|W_\perp(\boldsymbol{y} - \mu_{k-1})\|^2}{2(\sigma_k^\perp A_k^\perp)^2}\right)$$

$$= (2\pi)^{-d/2}\prod_{t=k}^T (\frac{B_t^{\mathcal{M}}}{A_t^{\mathcal{M}}})^{-m}(\frac{B_t^\perp}{A_t^\perp})^{m-d}(\sigma_k^{\mathcal{M}}A_k^{\mathcal{M}})^{-m}(\sigma_k^\perp A_k^\perp)^{m-d}$$

$$\exp\left(-\frac{\|W_{\mathcal{M}}(\boldsymbol{y} - \mu_{k-1})\|^2}{2(\sigma_k^{\mathcal{M}}A_k^{\mathcal{M}})^2} - \frac{\|W_\perp(\boldsymbol{y} - \mu_{k-1})\|^2}{2(\sigma_k^\perp A_k^\perp)^2}\right)$$

Adding zero-mean Gaussian white noise to $\boldsymbol{y}$ with variance $\sigma_k^2$ produces the desirable $\boldsymbol{s}_{k-1}$, which increase the variance in the original Gaussian bound by $\sigma_k^2$. Therefore, we can obtain

$$p_\theta(\boldsymbol{s}_{k-1}, k-1) \geq (2\pi)^{-d/2} \left( \prod_{t=k}^{T} (\frac{B_t^{\mathcal{M}}}{A_t^{\mathcal{M}}})^{-m} (\frac{B_t^\perp}{A_t^\perp})^{m-d} \right) (\sigma_k^2 + (A_k^{\mathcal{M}})^2(\sigma_k^{\mathcal{M}})^2)^{-m/2} (\sigma_k^2 + (A_k^\perp)^2(\sigma_k^\perp)^2)^{(m-d)/2}$$

(24)

$$\exp\left( -\frac{\|W_{\mathcal{M}}(\boldsymbol{s} - \mu_{k-1})\|^2}{2(\sigma_k^2 + (A_k^{\mathcal{M}})^2(\sigma_k^{\mathcal{M}})^2)} - \frac{\|W_\perp(\boldsymbol{s} - \mu_{k-1})\|^2}{2(\sigma_k^2 + (A_k^{\mathcal{M}})^2(\sigma_k^{\mathcal{M}})^2)} \right)$$

(25)

$$\geq (2\pi)^{-d/2} \left( \prod_{t=k}^{T} (\frac{B_t^{\mathcal{M}}}{A_t^{\mathcal{M}}})^{-m} (\frac{B_t^\perp}{A_t^\perp})^{m-d} \right) (\sigma_{k-1}^{\mathcal{M}})^{-m} (\sigma_{k-1}^\perp)^{m-d}$$

(26)

$$\exp\left( -\frac{\|W_{\mathcal{M}}(\boldsymbol{s} - \mu_{k-1})\|^2}{2(\sigma_{k-1}^{\mathcal{M}})^2} - \frac{\|W_\perp(\boldsymbol{s} - \mu_{k-1})\|^2}{2(\sigma_{k-1}^\perp)^2} \right)$$

(27)

Observe that since $\mu_T \in \mathcal{M}$ and $\mu_\theta$ is block preserving, $\mu_t \in \mathcal{M}$ for all $t$. we can simplify the expression to be

$$p_\theta(\boldsymbol{s}_{k-1}, k-1) \geq (2\pi)^{-d/2} \left( \prod_{t=k}^{T} (\frac{B_t^{\mathcal{M}}}{A_t^{\mathcal{M}}})^{-m} (\frac{B_t^\perp}{A_t^\perp})^{m-d} \right) (\sigma_{k-1}^{\mathcal{M}})^{-m} (\sigma_{k-1}^\perp)^{m-d} \exp\left( -\frac{\|W_{\mathcal{M}}(\boldsymbol{s} - \mu_{k-1})\|^2}{2(\sigma_{k-1}^{\mathcal{M}})^2} - \frac{\|W_\perp \boldsymbol{s}\|^2}{2(\sigma_{k-1}^\perp)^2} \right)$$

(28)

Hence we conclude the induction.

Let $C = (2\pi)^{-d/2} \left( \prod_{t=1}^{T} (\frac{B_t^{\mathcal{M}}}{A_t^{\mathcal{M}}})^{-m} (\frac{B_t^\perp}{A_t^\perp})^{m-d} \right) (\sigma_0^{\mathcal{M}})^{-m} (\sigma_0^\perp)^{m-d}$, we arrive at

$$p_\theta(\boldsymbol{s}) \geq C \exp\left( -\frac{\|W_{\mathcal{M}}(\boldsymbol{s} - \mu_0)\|^2}{2(\sigma_0^{\mathcal{M}})^2} - \frac{\|W_\perp \boldsymbol{s}\|^2}{2(\sigma_0^\perp)^2} \right)$$

(29)

$\square$

From this theorem above, we can make two observations: Firstly, the density assigned at a certain point depends on two factors: (1) how far it is along the manifold from the model's noise-free center, and (2) how far it is off the manifold. Secondly, since $0 < A_t^\perp \leq B_t^\perp < 1$, off-manifold components incurs much harsher penalty under the DDPM model, with likelihood dropping rapidly as the off-manifold component grows. In other words, a well-trained DDPM model concentrates on in-combination states, including the OOC ones. How much density it assigns to each in-combination state is determined by how far away the state is from the model learned noise-free center.

## C   Experiment Details

### C.1   Hardware and Platform

Experiments are run on a single NVIDIA RTX A6000 GPUs, with all code implemented in PyTorch.

### C.2   Statistics

All mean value is obtained by running with three different seeds and calculated with numpy.mean(). All error bar is obtained by numpy.std().

### C.3   Model Architecture

The backbone for Unet is based on Janner et al. (2022). We add cross-attention blocks after each residual block, except for the bottleneck layers. Inputs to the cross-attention blocks are the conditioning embedding and output of the residual block. To ensure local consistency of trajectory, we used 1D convolution along the horizon dimension. To keep the number of parameters for cross attention and the original Unet relatively balanced, we also used 1D convolution as the mapping from input to key, query, and value. Detailed model architecture is shown below in Figure 9.

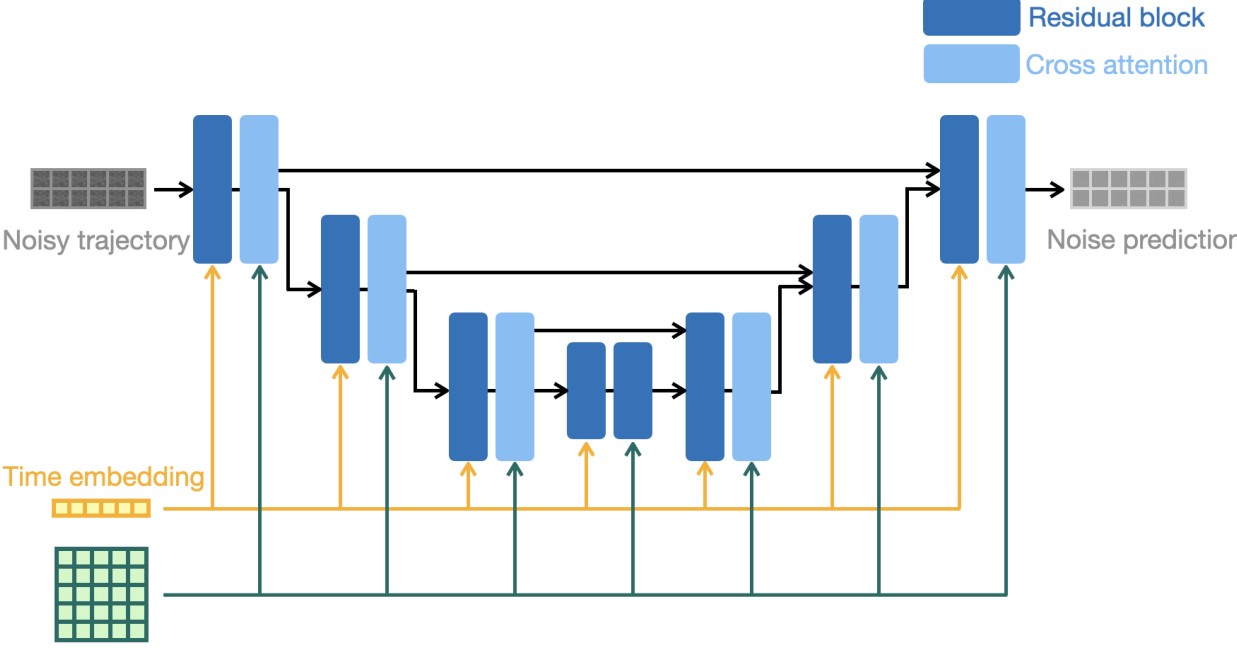

Figure 9: Model architecture.

The number of down-sampling/up-sampling and feature channel sizes is different for each experiment. Detailed parameters can be found in the section for each experiment.

### C.4 Trajectory formulation

The trajectory $\boldsymbol{\tau} \in \mathbb{R}^d$ is represented by concatenating the state $\boldsymbol{s_u} \in \mathbb{R}^{d_S}$ and the action $\boldsymbol{a_u} \in \mathbb{R}^{d_A}$ at planning time step $u$ and then horizontally stacking them for all time steps. For example, a trajectory with planning horizon $h$ can be written as $\boldsymbol{\tau} = \begin{bmatrix} \boldsymbol{s_1} & \boldsymbol{s_2} \ldots & \boldsymbol{s_h} \\ \boldsymbol{a_1} & \boldsymbol{a_2} \ldots & \boldsymbol{a_h} \end{bmatrix}$.

### C.5 Maze2D

### C.5.1 Extra Results

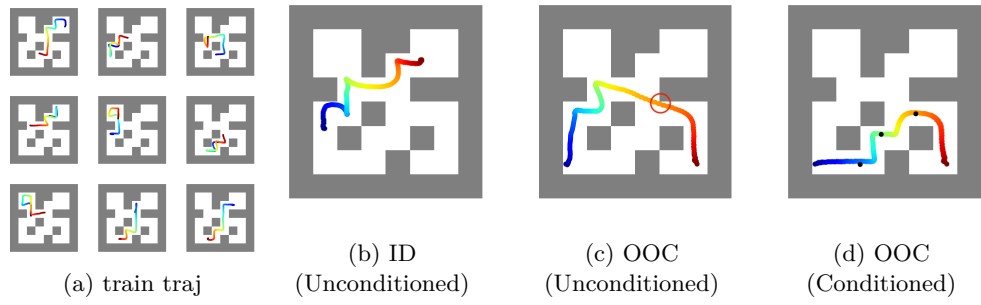

| (a) train traj | (b) ID (Unconditioned) | (c) OOC (Unconditioned) | (d) OOC (Conditioned) |

Figure 10: Trajectories generated in Maze2D for medium maze. (a) are samples from the training set. (b) are trajectories generated by the unconditioned diffusion model given in distribution start and end positions. (c) are generated by the unconditioned diffusion model on unseen start and end positions. (d) are generated by a conditioned diffusion model using 3 waypoints (black dots) as conditioning with classifier-free guidance (cfg) weight 1.3.

### C.5.2 Experiment Details

We followed the setup used in Janner et al. (2022). The hyperparameters shared for large and medium mazes are shown below in Table 1. Large maze use a planning horizon of 384 and medium maze use a planning horizon of 256. Conditioning is passed through a positional embedding layer first to map each dimension of the waypoint $(x, y, v_x, v_y)$ to a higher dimension of 21 and concatenate them to form a vector of size $(1, 21 * 4)$. Three waypoints are then stacked together to form a matrix of size $(3, 21 * 4)$ and passed into the cross-attention layer. In our experiment, directly using the waypoints as conditioning was unsuccessful.

| Parameter | Value |
|---|---|
| number of diffusion steps | 256 |
| action weight | 1 |
| dimension multipliers | (1, 4, 8) |
| classifier free guidance drop conditioning probability | 0.1 |
| steps per epoch | 10000 |
| loss type | l2 |
| train steps | 2e6 |
| batch size | 32 |
| learning rate | 2e-4 |
| gradient accumulate every | 2 |
| ema decay | 0.995 |

Table 1: Training parameter for diffusion model in Maze2D

### C.6 Roundabout

#### C.6.1 Environment

The training environment consists of all cars or all bicycles and the testing environment is a mixture of traffic (Figure 11).

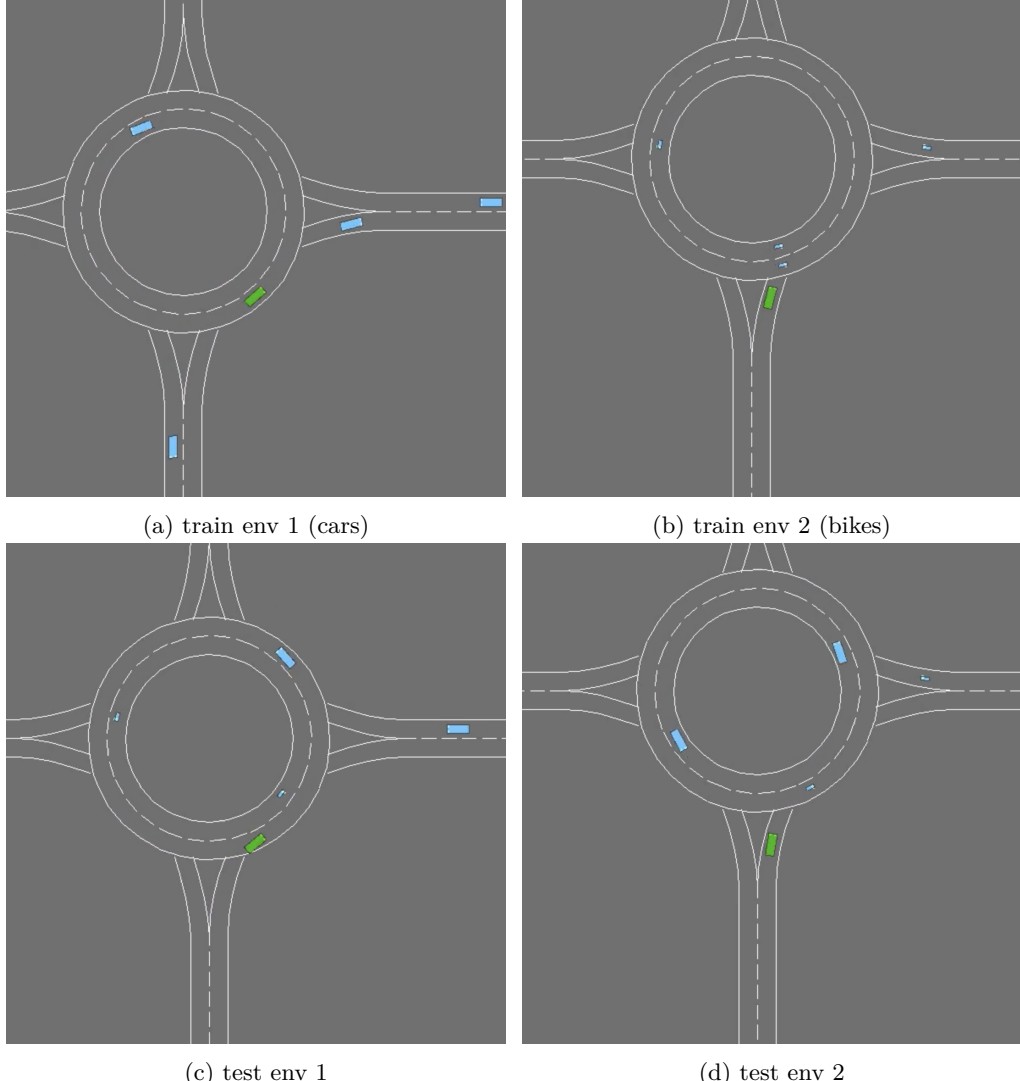

(a) train env 1 (cars)                    (b) train env 2 (bikes)

(c) test env 1                            (d) test env 2

Figure 11: Training and testing environments for Roundabout. The green vehicle is the ego agent and the blue ones are controlled by the environment. The large blue box represents a car and the small blue box represents a bicycle.

#### C.6.2 Environment Parameters

We changed the parameters to create a different type of traffic in the roundabout as shown below in Table 4. Also, since bicycles have slower speeds, we change the initialization position so that each environment vehicle can interact with the ego vehicle.

| Parameter | Car | Bicycle |
|---|---|---|
| length | 5.0 | 2.0 |
| width | 2.0 | 1.0 |
| speed | [23, 25] | 4 |
| max acceleration | 6.0 | 2.0 |
| comfort max acceleration | 3.0 | 1.0 |

Table 2: Parameters for car and bicycles in Roundabout environment

### C.6.3 Dataset

In order to collect expert trajectories, we train two PPO models separately on the environment with all cars and all bicycles. We then collect 320000 successful trajectories in the training environment. All trajectories have a unified length of 12.

## C.7 Setup for Roundabout Environment

Here we describe how the Roundabout task in this paper conforms to our problem description. In this setting our base object set is $E = \{car, bicycle, null\}$ where null means an object is non-visible. Since the maximum number of objects in the roundabout is five and we fix the ego agent to be a car, support for the training observation is $\{(car\ (ego\ agent), car, car, car, car), (car\ (ego\ agent), bicycle, bicycle, bicycle, bicycle)\}$ and for the testing observation is $\{(car\ (ego\ agent), bicycle, bicycle, car, car)\}$ assuming no ordering and when the state is fully observable. Since the supports for training and testing are non-overlapping under full observability, they will remain non-overlapping even when some traffic objects are out of sight, unless the ego agent is the only object present in the environment.

### C.7.1 Experiment Details

We use stable_baseline3 Raffin et al. (2021) as the implementation for PPO. The parameter is the default parameter used in the Highway environment Leurent (2018). We increased total timesteps because the environment now has two modalities (all cars and all bicycles) and we observed that PPO takes longer to converge. Detailed parameters for PPO and diffusion are shown below in Table 3 and Table 4.

| Parameter | Value |
|---|---|
| policy | MlpPolicy |
| batch size | 64 |
| n_steps | 768 |
| n_epochs | 10 |
| learning rate | 5e-4 |
| gamma | 0.8 |
| total timesteps | 2e5 |

Table 3: Training parameter for PPO

| Parameter | Value |
|---|---|
| planning horizon | 8 |
| number of diffusion steps | 80 |
| action weight | 10 |
| dimension multipliers | (1, 4, 8) |
| conditioning embedding size | 20 |
| classifier free guidance drop conditioning probability | 0.1 |
| classifier free guidance weight | 1.0 |
| steps per epoch | 10000 |
| loss type | l2 |
| train steps | 1e4 |
| batch size | 32 |
| learning rate | 2e-4 |
| gradient accumulate every | 2 |
| ema decay | 0.995 |

Table 4: Training parameter for diffusion model in Roundabout

### C.7.2 Model Size

We include the model size for different algorithms below in Table 5. To eliminate the concern for performance gain due to model size, we include the performance of a large BC model that has roughly the same number of parameters as the conditioned diffusion model.

| | BC | PPO | Diffusion | Large BC |
|---|---|---|---|---|
| Model size | 0.30 MB | 0.60 MB | 54.19 MB | 55.65 MB |
| Number of parameters | 75013 | 148998 | 13546370 | 13912325 |
| OOD reward | 7.50 (0.03) | 8.19 (0.16) | 8.81 (0.2) | 7.71 (0.3) |
| OOD crashes | 37.7 (0.5) | 36.0 (5.7) | 20.0 (2.5) | 37.3 (4.0) |

Table 5: Model size, number of parameters, and performance for different models.

### C.8 StarCraft

### C.8.1 Experiment Details

We use the codebase OpenRL Huang et al. (2023) for the implementation of MAPPO. Detailed parameters for MAPPO can be found in Table 6.

| Parameter | Value |
|---|---|
| learning rate actor | 5e-4 |
| learning rate critic | 1e-3 |
| data chunk length | 8 |
| env num | 8 |
| episode length | 400 |
| PPO epoch | 5 |
| actor train interval step | 1 |
| use recurrent policy | True |
| use adv normalize | True |
| use value active masks | False |
| use linear LR decay | True |

Table 6: MAPPO hyper-parameters used for SMACv2. We utilize the hyperparameters used in SMACv2 Ellis et al. (2022).

Detailed parameters for training a conditioned diffusion model for 5v5 are shown below in Table 7 and 3v3 in Table 8.

| Parameter | Value |
|---|---|
| planning horizon | 40 |
| number of diffusion steps | 256 |
| action weight | 1 |
| dimension multipliers | (1, 4, 8) |
| conditioning embedding size | 40 |
| classifier free guidance drop conditioning probability | 0.1 |
| classifier free guidance weight | [0.7, 1.0, 1.3, 1.5] |
| steps per epoch | 10000 |
| loss type | l2 |
| train steps | 2e6 |
| batch size | 32 |
| learning rate | 2e-4 |
| gradient accumulate every | 2 |
| ema decay | 0.995 |

Table 7: Training parameter for diffusion model in StarCraft for 5v5

| Parameter | Value |
|---|---|
| planning horizon | 32 |
| number of diffusion steps | 256 |
| action weight | 1 |
| dimension multipliers | (1, 4, 8) |
| conditioning embedding size | 40 |
| classifier free guidance drop conditioning probability | 0.1 |
| classifier free guidance weight | [0.7, 1.0, 1.3, 1.5] |
| steps per epoch | 10000 |
| loss type | l2 |
| train steps | 2e6 |
| batch size | 32 |
| learning rate | 2e-4 |
| gradient accumulate every | 2 |
| ema decay | 0.995 |

Table 8: Training parameter for diffusion model in StarCraft for 3v3

### C.8.2 Dataset Initial State Distribution

The probability of generating each unit type in SMACv2 is imbalanced. Specifically, the probability for Stalker, Zealot, and Colossus is 0.45, 0.45, and 0.1 respectively. The initial state distribution of training trajectories collected by MAPPO for random combination is shown below in Figure 12a and 12b. Since we only keep the successful trajectories and use them as expert data, the distribution depends on the generation probability and MAPPO success rate for different team combinations. A total number of 240000 trajectories were used to train the diffusion model. Since diffusion is trained on local observations and actions of all MAPPO actors, the total number of training samples is 5*240000 for 5v5 and 3*240000 for 3v3.

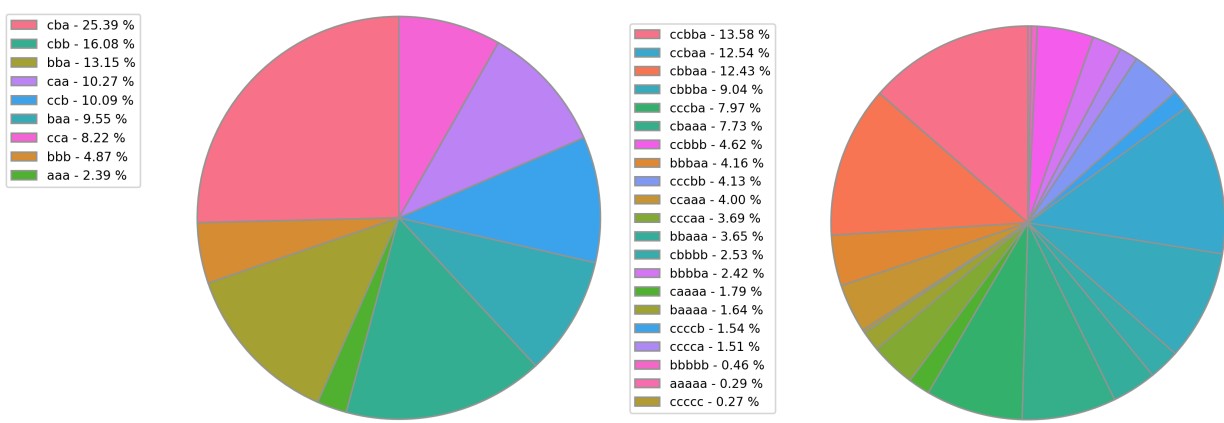

(a) Distribution of initial state for 3v3 simple scenario    (b) Distribution of initial state for 5v5 simple scenario

### C.8.3 Detailed Results on SMACv2

Table 9 and 10 show the detailed performance of different algorithms in the 3v3 and 5v5 scenarios, respectively.

| Env: 3v3 | RL | | Imitation Learning | |
|---|---|---|---|---|
| | 2 PPO + 1 Rand | 3 PPO | BC | 2 PPO + 1 Diffusion |
| $ABC \rightarrow ABC$ (ID) | 0.18 (0.01) | 0.58 (0.02) | 0.58 (0.07) | 0.59 (0.04) |
| $ABC \rightarrow AAA$ (Simple) | 0.07 (0.03) | 0.52 (0.03) | 0.56 (0.02) | 0.59 (0.02) |
| $AAA \rightarrow AAA$ (ID) | 0.09 (0.04) | 0.63 (0.02) | 0.6 (0.02) | 0.61 (0.05) |
| $AAA \rightarrow ABC$ (Hard) | 0.11 (0.02) | 0.42 (0.02) | 0.4 (0.06) | 0.49(0.02) |

Table 9: Success rate of each agent in 100 rounds. The first two rows correspond to the simple setting of generalization to states with different support and the last two rows correspond to non-overlapping support. Numbers in the parenthesis represent the standard error over 3 seeds. The best performing method is labeled with blue color. The 2 PPO + 1 Rand column shows the effect of replacing one PPO trained agent with a random agent as a baseline for comparison against the 2 PPO + 1 Diffusion case.

| Env: 5v5 | RL | | Imitation Learning | |
|---|---|---|---|---|
| | 4 PPO + 1 Rand | 5 PPO | BC | 4 PPO + 1 Diffusion |
| $ABC \rightarrow ABC$ (ID) | 0.22 (0.04) | 0.64 (0.05) | 0.56 (0.05) | 0.66 (0.01) |
| $ABC \rightarrow AAA$ (Simple) | 0.11 (0.03) | 0.54 (0.04) | 0.52 (0.05) | 0.56 (0.02) |
| $AAA \rightarrow AAA$ (ID) | 0.14 (0.02) | 0.58 (0.04) | 0.54 (0.04) | 0.55 (0.03) |
| $AAA \rightarrow ABC$ (Hard) | 0.11 (0.02) | 0.26 (0.05) | 0.28 (0.04) | 0.32 (0.04) |

Table 10: Success rate of each agent in 100 rounds. The first two rows correspond to the simple setting of generalization to states with different support and the last two rows correspond to non-overlapping support. Numbers in the parenthesis represent the standard error over 3 seeds. The best performing method is labeled with blue color. The 4 PPO + 1 Rand column shows the effect of replacing one PPO trained agent with a random agent as a baseline for comparison against the 4 PPO + 1 Diffusion case.

### C.8.4 Detailed Results for Ablation

The ablation result for 3v3 and 5v5 scenarios are shown below in Table 11 and Table 12. The first column is the success rate without conditioning (No Cond). The second column represents concatenating the conditioning with time embedding (Concat). The last column represents passing conditioning as another input beside the trajectory to the cross-attention block (Attn).

Table 11: Ablation for Diffusion on 3v3

| Env 3v3 | 2 PPO + 1 Diffusion | | |
|---|---|---|---|
| | No Cond | Concat | Attn |
| $ABC \rightarrow ABC$ (ID) | 0.55±0.03 | 0.59±0.04 | 0.59±0.05 |
| $ABC \rightarrow AAA$ (Simple) | 0.5±0.06 | 0.59±0.02 | 0.59±0.02 |
| $AAA \rightarrow AAA$ (ID) | 0.4±0.03 | 0.64±0.03 | 0.61±0.05 |
| $AAA \rightarrow ABC$ (Hard) | 0.28±0.03 | 0.44±0.05 | 0.49±0.02 |

Table 12: Ablation for Diffusion on 5v5

| Env 5v5 | 4PPO + 1 Diffusion | | |
| --- | --- | --- | --- |
| | No Cond | Concat | Attn |
| $ABC \to ABC$ (ID) | 0.53±0.04 | 0.59±0.03 | 0.66±0.01 |
| $ABC \to AAA$ (Simple) | 0.50±0.03 | 0.50±0.01 | 0.56±0.02 |
| $AAA \to AAA$ (ID) | 0.47±0.08 | 0.55±0.03 | 0.58±0.04 |
| $AAA \to ABC$ (Hard) | 0.27±0.03 | 0.32±0.04 | 0.30±0.04 |

### C.8.5 2v2

The success rates for StarCraft 2v2 are shown below in Table 13. We can see that out-of-combination cases did not cause the performance to drop drastically for MAPPO. This is because the number of combinations in 2v2 is very limited (e.g. $aa, bb, ab$), and if one agent dies, MAPPO has encountered scenarios of playing with each unit type individually, therefore falling back to in distribution state again. This scenario also exists for 5v5 and 3v3 but only at the end of each game when only one agent is left.

Table 13: SMAC II success rate for 2v2

| Env | BC | MAPPO | | Diffusion | | |
| --- | --- | --- | --- | --- | --- | --- |
| | BC | 1 PPO + 1 Rand | 5 PPO | No Cond | Concat | Attn |
| $ABC \to ABC$ (ID) | 0.54±0.02 | 0.06±0.02 | 0.62±0.05 | 0.43±0.04 | 0.56±0.03 | 0.57±0.067 |
| $ABC \to AAA$ (Simple) | 0.47±0.02 | 0.02±0.01 | 0.57±0.02 | 0.44±0.02 | 0.55±0.06 | 0.49±0.06 |
| $AAA \to AAA$ (ID) | 0.57±0.01 | 0.01±0.01 | 0.64±0 | 0.38±0.02 | 0.63±0.04 | 0.63±0.02 |
| $AAA \to ABC$ (Hard) | 0.4±0.03 | 0.04±0.02 | 0.44±0.02 | 0.29±0.02 | 0.43±0.08 | 0.41±0.06 |

### C.8.6 More Rendering of States Predicted by the Diffusion model

More rendering of the future states predicted by the diffusion model is shown in Figure 13.

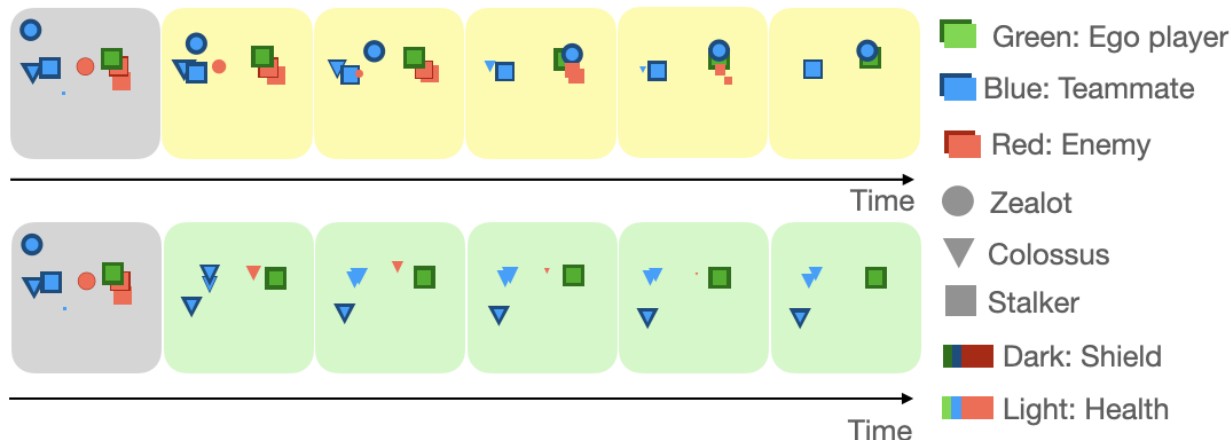

Figure 13: Rendering of future states predicted by the diffusion model given different conditionings. The grey box is the initial state. Yellow boxes are conditioned on the type of unit in the initial state. Green boxes are conditioned on all Triangles. Smaller sizes represent less shield or health.

### C.9 Model Runtime and GPU Memory

We include the training time and GPU memory used for the conditioned diffusion model below in Table 14 and 15.

| Training Time | **Roundabout** | **SMACv2 2v2** | **SMACv2 3v3** | **SMACv2 5v5** |
|---|---|---|---|---|
| PPO | 0.5h | 9h | 9h | 9h |
| Diffusion | 1h | 48h | 70h | 98h |

Table 14: Training time for PPO and conditioned diffusion model in different environments.

| GPU Memory | **Roundabout** | **SMACv2 2v2** | **SMACv2 3v3** | **SMACv2 5v5** |
|---|---|---|---|---|
| Diffusion | 542 MiB | 1004 MiB | 2892 MiB | 4096 MiB |

Table 15: GPU Memory for training conditioned diffusion model in different environments.

### C.9.1 Substituting More MAPPO Agents with Diffusion Agents

We would like to ask the question of what about replacing more than one MAPPO agent with diffusion model. Figure 14 shows that the number of diffusion models does not have a positive correlation with the success rate. This is because MAPPO can learn a collaborative policy between actors and simply adding more ego-centric diffusion models will break the coordination between actions. Also, since the diffusion model is trained to play with all PPOs teammates, replacing other PPO actions with actions generated by diffusion models will cause a distribution shift that is hard to quantify.

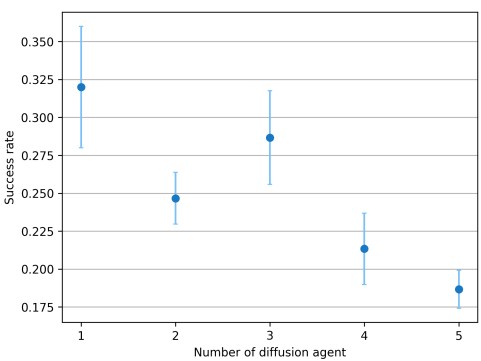

Figure 14: Success rate vs number of agents in SMACv2 5v5 hard scenario that are replaced with diffusion agents. Replacing more than one MAPPO agent with diffusion agents hurts performance.

