# OpenReview forum: "State Combinatorial Generalization In Decision Making With Conditional Diffusion Models"
_TMLR — Accepted by TMLR_

### Review · Reviewer_1rA2 · 2025-09-09

**Summary Of Contributions:**

This paper investigates decision-making problems in reinforcement learning where states are composed of basic elements. Building on this perspective, the authors introduce the state combinatorial generalization problem, where agents must handle novel combinations of known elements at test time. They show that traditional RL approaches struggle with such out-of-support generalization due to unreliable value predictions. To address this, the authors propose using conditional diffusion models, leveraging their ability to capture compositional structure. Specifically, they define a structured vector Z representing element combinations and use conditioning on Z to guide trajectory generation. The diffusion model is trained offline on expert demonstrations but deployed online for decision-making. Empirical evaluations across diverse environments, including driving (Roundabout), maze navigation (Maze2D), and multi-agent games (SMACv2), demonstrate that the proposed method consistently outperforms both offline and online RL baselines in zero-shot generalization tasks.

**Audience:**

Yes

**Audience Explanation:**

The paper provides a strong motivation for studying generalization in decision-making tasks, particularly focusing on handling unseen combinations of known elements. The concept of state combinatorial generalization and the out-of-combination (OOC) are clearly defined and distinguished from traditional out-of-distribution (OOD) generalization problems.

The proposed approach demonstrates impressive results, showing that a diffusion model trained offline via behavior cloning can outperform strong online RL baselines such as PPO in zero-shot OOC generalization tasks.

**Broader Impact Concerns:**

No significant concerns.

**Claims And Evidence:**

Yes

**Claims Explanation:**

Extensive experiments are conducted to cover a diverse range of environments. Detailed ablation studies and visualizations are also conducted to help clarify why the diffusion model generalizes effectively.

**Requested Changes:**

Major issue:

To explain why diffusion models generalize better, the paper introduces Corollary 5.1, which shows that a well-trained diffusion model can, in principle, sample out-of-combination (OOC) instances with non-zero probability. Later, in Section 6, the authors argue that the generated trajectories will remain dynamically consistent when conditioned on novel combinations of base elements. While both points are interesting and seem to be loosely connected, the connection between them is not sufficiently clear. Specifically, it is unclear how the existence result in Corollary 5.1 translates into the practical ability to handle unseen conditioning vectors in a reliable way. For example, if the probability of sampling an OOC instance is technically non-zero but extremely small, how does conditioning ensure that such states are generated with meaningful likelihood? A more rigorous theoretical link between the corollary and the conditioning-based generalization claim would significantly strengthen the argument.

Minor issue:

While Section 6 provides a description of the training and testing procedure for the proposed diffusion model, the explanation could be made clearer and more structured. A step-by-step outline or schematic would help readers better understand how the model is trained offline and then deployed online for decision-making.

The terminology around Z is potentially confusing. Referring to it as a latent vector may be misleading, since it is explicitly defined to represent the combination of base elements and is therefore not latent in the conventional sense. A more precise term, such as compositional encoder (similar to positional encoder in transformers), might avoid confusion.

The proposed method generates full trajectories during inference, but only the first action is executed in the environment. This raises a potential computational efficiency concern, as much of the generated trajectory is discarded at each step. A discussion of potential strategies to mitigate it would be useful.

---

> ### Author Response · Authors · 2025-10-02
>
> **1. How the existence result in Corollary 5.1 translates into the practical ability to handle unseen conditioning vectors in a reliable way:** We thank the reviewer for highlighting this important point and have expanded our analysis accordingly.
>
> In Appendix B, we now provide a refined theoretical characterization of when diffusion models assign non-negligible density to OOC states. Specifically, we establish a lower bound on the density of a sample, showing that likelihood depends on two factors: (1) its distance along the manifold from the model’s noise-free center, and (2) its distance off the manifold. A well-trained DDPM thus concentrates density on valid in-combination states (seen and OOC) and assigns non-negligible mass to unseen but valid combinations (OOC states), while strongly suppressing invalid or out-of-manifold states. Whether an OOC state is generated with practical frequency depends on its proximity to the model’s learned noise-free center, which aligns well with empirical findings in CV literature[1].
>
> This theoretical analysis captures the possibility of generating OOC samples, but in practice, reliable generalization requires a well-trained diffusion model. Prior computer vision studies (Section 5.2) show that **conditioning** significantly improves sampling outside the training support. Consistently, our experiments confirm this: in Section 8.1 (Fig. 7), unconditioned diffusion models succeed only when both start and goal locations were seen during training, whereas conditioned models generalize to novel start–goal combinations when provided with novel waypoints as conditioning. Similarly, in Section 7.3, conditioning enables accurate trajectory generation in multi-agent settings with unseen player type combinations.
>
> Together, these results establish a clearer link: the theoretical analysis ensures OOC states can, in principle, be assigned non-negligible density, while experimental results demonstrate that conditioning can guide the model in practice by shifting probability density toward relevant OOC states under novel conditioning vectors.
>
>
> [1]Okawa, Maya, et al. "Compositional abilities emerge multiplicatively: Exploring diffusion models on a synthetic task." Advances in Neural Information Processing Systems 36 (2023): 50173-50195.
>
>
> **2. A step-by-step outline for model training and deployment for decision-making.** We really appreciate your suggestion and we have moved the pseudo-code outlining the step-by-step procedure of how to use conditioned diffusion as a planner to the main text.
>
> **3. The terminology around Z:** Thanks for pointing out this confusion! We propose to change the name to compositional vector in the camera-ready version. We are keeping it as the current version to avoid misreference during discussion.
>
> **4. computational efficiency concern:** We thank the reviewer for pointing out this limitation. Various approaches have been proposed to accelerate the sampling process in diffusion models via advanced ODE solvers or knowledge distillation techniques that could be applied to mitigate the issue of time efficiency. We have added the above discussion in the updated version.

---

### Review · Reviewer_fsvP · 2025-09-17

**Summary Of Contributions:**

The authors study 'out-of-combination' (OOC) generalisation, an OOD setting where new combinations of concepts are introduced. All concepts are however individually shown during training. Then the authors study this in an RL setting (sort of), and argue that diffusion models generalise better to this OOC setting. They experiment with several settings and show that conditioning on latent concepts improves performance in diffusion models.

Strengths:
- Important and timely topic
- Writing is quite good

Weaknesses:
- Not really explored in a decision-making context
- Evaluation is on pretty artificial datasets
- Theoretical arguments for diffusion are unconvincing

**Additional Comments:**

- In algorithm 1, why is the state o noised instead of just conditioning directly on the observation o?
- Is it correct the setting in Figure 1 is not used in any experiments
- I did not understand the experiment of 8.1. How is the OOC setting differently constructed? And how is z represented?

**Audience:**

Yes

**Audience Explanation:**

The topic of the paper certainly is of interest to a fairly broad subset.

**Claims And Evidence:**

No

**Claims Explanation:**

- The theoretical argument in Corollary 5.1 neither seems relevant nor correct
- The setup in the experiments is confusing and complicated to me. It assumes behaviour cloning of an existing PPO-trained agent. Then, I think it uses a hard-coded concept extractor to extract the concepts from the image. This is usually not available in real-world scenarios.
- The paper should be clearer about its experimental setup and methodology, including all the steps to get to the final model(s).
- The use of diffusion to improve combinatorial generalisation is not well supported. In particular, there does not seem to be any experiments with conditioned non-diffusion models.

**Requested Changes:**

Critical
- I fail to follow the point of Corollary 5.1, which is the main theoretical motivation for 'OOC Generalization in Diffusion Models'. The authors should clarify and improve the theoretical backing for diffusion models.
    - In my understanding, the authors argue that a diffusion model will sample some state $s'$ lying in a linear manifold with non-zero probability under a diffusion model.
    - This is incorrect: The probability of sampling a particular state within a continuous manifold is almost surely 0. I suppose the authors meant the density is nonzero.
    - If so: This doesn't say much. Any Gaussian will have the same result, it's just saying $s'$ is in the support.
    - Furthermore, this statement has nothing to do with $s'$ being OOC or not.
- The authors should compare to a BC method without diffusion models but with conditioning on $z$. This seems to be an important factor contributing to the methods success, but is not properly ablated.
- The authors should highlight that they assume the latent concept $z$ is given to the learner. This is not usually assumed in RL.
- The conditioning method is not clear to me. Some points mention just conditioning the model, but then for the maze experiment, classifier-free guidance is used.
- It was not clear how the diffusion model is trained to deal with $z$. Also, is the PPO agent trained with knowledge of $z$?

Enhancements
- Probably it's clear to most, but I would emphasise more that OOC is a specific type of OOD generalisation
- Relevant related work:
    - Combinatorial generalisation is also frequently studied within NLP, eg [1, 2], although this is possibly harder to consider as the formal OOC setup of this paper.
    - The setup in Section 3 is closely related to that studied in Neurosymbolic Learning [3], both formally and informally. Some settings in that paper are examples of OOC (eg Mnist Even-Odd). On such datasets, it was also recently shown that diffusion can help generalisation [4]. One difference is that this line of work does not assume the latent vector of concepts z are given, but are extracted with neural networks. It is also related to concept-based models where concepts z are only given at training time [5].
- 5.1: Diffusion is not used for 'density estimation', as it does not have tractable density computation (one can only sample from it)



1. Hupkes, Dieuwke, et al. "Compositionality decomposed: How do neural networks generalise?." Journal of Artificial Intelligence Research 67 (2020): 757-795.
2. Press, Ofir, et al. "Measuring and narrowing the compositionality gap in language models." arXiv preprint arXiv:2210.03350 (2022).
3. Marconato, Emanuele, et al. "Not all neuro-symbolic concepts are created equal: Analysis and mitigation of reasoning shortcuts." Advances in Neural Information Processing Systems 36 (2023): 72507-72539.
4. van Krieken, Emile, et al. "Neurosymbolic Diffusion Models." arXiv preprint arXiv:2505.13138 (2025).
5. Kim, Been, et al. "Interpretability beyond feature attribution: Quantitative testing with concept activation vectors (tcav)." International conference on machine learning. PMLR, 2018.

---

> ### Author Response · Authors · 2025-10-02
>
> We sincerely thank the reviewer for the thoughtful comments and constructive suggestions, which have helped us clarify our theoretical analysis and improve the overall presentation of the paper. We will address the concerns below:
>
> **1. Theoretical justification:** We would like to sincerely thank the reviewers for raising the concerns related to the theoretical analysis. We agree that our old version did not characterize the density clearly, and after incorporating your suggestions, we provide a clearer theoretical characterization of generalization in Appendix B. Specifically, we establish a lower bound on the density of a sample, showing that likelihood depends on two factors: (1) its distance along the manifold from the model’s noise-free center, and (2) its distance off the manifold. A well-trained DDPM thus concentrates density on valid in-combination (seen and OOC) states and assigns non-negligible mass to unseen but valid combinations (OOC states), while strongly suppressing invalid or out-of-manifold states. Whether an OOC state is generated with practical frequency depends on its proximity to the model’s learned noise-free center, which aligns well with empirical findings in CV literature[1].
>
> [1]Okawa, Maya, et al. "Compositional abilities emerge multiplicatively: Exploring diffusion models on a synthetic task." Advances in Neural Information Processing Systems 36 (2023): 50173-50195.
>
> **2. Diffusion is not used for 'density estimation', as it does not have tractable density computation (one can only sample from it):** We would like to respectfully point out that under the probability flow ODE formulation of diffusion models, exact likelihood can indeed be computed, as shown by Song et al. [2].
>
> [2]Song, Yang, et al. "Score-based generative modeling through stochastic differential equations." arXiv preprint arXiv:2011.13456 (2020)
>
>
> **3. BC without diffusion models conditioned on z; Is PPO trained with z:** We would like to clarify that our reported results of BC/PPO already correspond to *conditioned* behavior cloning. Since z is included in the observation space, it is available to all models, meaning every baseline is conditioned on z. The difference lies only in how this information is incorporated: conditioned diffusion uses classifier-free guidance, while other methods leverage it directly through the observation input.
>
> **4. Conditioning method:** We would like to clarify that classifier-free guidance is the method we consistently use to incorporate compositional conditioning into diffusion models throughout the paper, including the maze experiment. We will revise the text to make this explicit.
>
>
> **5. How the diffusion model is trained to deal with z:** We thank the reviewer for this suggestion and have moved Algorithm 1 into the main text. In conditioned diffusion training, we separate z from the rest of the observation space, and the model is trained to denoise future trajectories conditioned on both the current observation and z. We follow the same setup as Diffuser and Conditioned Diffuser [3,4], where the current observation is enforced by swapping the first observation in each intermediate denoising trajectory with the true observed state at every timestep. Conditioning on z is incorporated through classifier-free guidance.
>
> [3]Janner, Michael, et al. "Planning with diffusion for flexible behavior synthesis." arXiv preprint arXiv:2205.09991 (2022).
>
> [4]Ajay, Anurag, et al. "Is conditional generative modeling all you need for decision-making?." arXiv preprint arXiv:2211.15657 (2022).
>
>
> **6. Concept extractor for image:** We would like to clarify that all of our environments are state-based rather than image-based. Information such as traffic type, player type, and waypoint locations is explicitly provided in the environment observation. For conditioned diffusion, we simply separated this information from the rest of the observation, rather than relying on a concept extractor.
>
>
> **7. Related work:** We sincerely appreciate your clarification suggestions and related work mentioned in the field of NLP and neurosymbolic learning! We will incorporate those discussions in the camera-ready version.
>
>
> **8. noised o in algorithm 1:** Thank you so much for pointing out this mistake. We have updated Algorithm 1 to: Replace the first denoised state with observed o.
>
> **9. Is the setting in Figure 1 used in any experiments?** Figure 1 is intended as a high-level conceptual illustration of the SMACv2 multi-agent game environment.
>
> **10. Setup for experiment 8.1:** Unlike the previous environments, we formalize this as a single-step planning problem rather than a multi-step one. The entire trajectory is generated at once and then executed without replanning. State conditioning corresponds to the observed start and goal locations, while compositional conditioning corresponds to different waypoint combinations that the trajectory must pass through.

---

### Review · Reviewer_qK9o · 2025-09-18

**Summary Of Contributions:**

This paper studies out of combination (OOC) generalisation, i.e. generalisation in RL to states that contain new combinations of elements not seen during training. This is not so much a problem of distributional shift, but rather of the support of the distribution shifting towards new combinations. The paper shows that Q-learning struggles here because OOC states are not encountered during training, meaning that more training will not help. On the contrary, the paper shows that diffusionn models do have a non-zero probability of sampling OOC states. Planning with a diffusion model should therefore work better. Experimentally this is confirmed.

**Audience:**

Yes

**Audience Explanation:**

Out-of-support generalization, as a complement to the many existing works on distributional shifts, is an important problem to study, and without a doubt of interest to RL researchers and ML researchers more generally. A comparison of the generalisation abilities within this setting of various RL and behavioural cloning methods is a valuable contribution.

**Broader Impact Concerns:**

No special concerns.

**Claims And Evidence:**

No

**Claims Explanation:**

I think the empirical side of this paper is generally too weak. The paper aims to show that diffusion models generalize better on out of combination states than classical behavioral cloning. But this is done on only two environments, with only one approach to behavioral cloning. Though its message is an interesting one, for this message to carry weight, evaluation would have to consider more environments and more baselines.

**Requested Changes:**

Critical to ensuring my recommendation would be first and foremost to extend the empirical evaluation to cover a wider range of environments. I would also like to see a better characterization of when diffusion models generalize well to OOC states and when they do not. Theoretically, we know that these states will be samples with non-zero probability, but as you say in the paper, the sampling probability can still be very near zero, meaning that practically generalisation is not guaranteed.

---

> ### Author Response · Authors · 2025-10-02
>
> **1. More baselines:** We thank the reviewer for suggesting additional baselines and environments. We have expanded our comparisons and the results in the updated paper now include:
>
> * Highway: Behavior cloning baselines (vanilla BC and conditioned diffusion) as well as reinforcement learning methods: PPO (online) and CQL (offline).
>
> * SMACv2: Behavior cloning methods (vanilla BC and conditioned diffusion), reinforcement learning methods (PPO (online and used as baseline for relative performance improvement), BCQ (offline), and CQL (offline)), and a random policy for reference.
>
> * Maze2D: Presented qualitative comparisons by visualizing trajectories generated from conditioned and unconditioned diffusion models.
>
> These three environments collectively cover single-agent, multi-agent, and long-horizon planning tasks, illustrating our model’s performance across diverse settings.
>
>
>
> **2. When diffusion models generalize well and when they do not:** We thank the reviewer for highlighting this important point and have expanded our analysis accordingly.
>
> In Appendix B, we provide a clearer theoretical characterization of generalization. Specifically, we establish a lower bound on the density of a sample, showing that likelihood depends on two factors: (1) its distance along the manifold from the model’s noise-free center, and (2) its distance off the manifold. A well-trained DDPM thus concentrates density on valid in-combination states (seen and OOC) and assigns non-negligible mass to unseen but valid combinations (OOC states), while strongly suppressing invalid or out-of-manifold states. Whether an OOC state is generated with practical frequency depends on its proximity to the model’s learned noise-free center, which aligns well with empirical findings in CV literature[1].
>
> This theoretical analysis captures the possibility of generating OOC samples, but in practice, reliable generalization requires a well-trained diffusion model. Prior computer vision studies (Section 5.2) show that **conditioning** significantly improves sampling outside the training support. Consistently, our experiments confirm this: in Section 8.1 (Fig. 7), unconditioned diffusion models succeed only when both start and goal locations were seen during training, whereas conditioned models generalize to novel start–goal combinations when provided with novel waypoints as conditioning. Similarly, in Section 7.3, conditioning enables accurate trajectory generation in multi-agent settings with unseen player type combinations.
>
> Together, these results clarify the conditions under which diffusion models will assign non-negligible density to OOC states, and effective yet reliable generalization is achieved when using conditioning to guide the sampling process.
>
> [1]Okawa, Maya, et al. "Compositional abilities emerge multiplicatively: Exploring diffusion models on a synthetic task." Advances in Neural Information Processing Systems 36 (2023): 50173-50195.

---

### Comment · Action_Editor_N3Y6 · 2025-10-31
**Clarification Needed**

Dear authors,

A clarification is needed on this paper.

Several of the reviewers brought up the problem with Corollary 5.1 and the biggest concern right now is that the phrasing "nonzero" was changed to "non-negligible" with some new proofs in the appendix.

Corollary 5.1 is not formal enough in its current form. Specifically, "non-negligible" is vague. A theoretical statement in the main paper that is listed as a main contribution needs to be more formal. Also, there should be a proof sketch or, at the very least, and informal version of the assumptions required for the proofs, stated in the main paper.

However, please understand: the critical part is defining very precisely what is meant by "non-negligible" here (which must be attached to the theorem statement in the main text). There is a big gap between the statement of Corollary 5.1 in the main paper and Equations (6) and (7) in the appendix. There are two paragraphs at the bottom of page 21 that discuss this, but do not clarify the claim that the lower-bound is "non-negligible" (in fact, it makes the same statement without clarifying what counts as negligible and what counts as non-negligible).

Please update the paper within the next 7 days to clarify this point.

---

> ### Comment · Action_Editor_N3Y6 · 2025-10-31
>
> If you can no longer update the paper, then please clarify the statement here in a response to this comment.

---

> > ### Author Response · Authors · 2025-11-05
> >
> > Thank you for the thoughtful and constructive feedback from the editor and reviewers. We truly appreciate the time and care that went into identifying areas for clarification. In the revised PDF, we have replaced “non-negligible” with a precise, quantified lower bound, moved the key assumptions, final bound, and proof sketch into the main text, and aligned the statement of Theorem 5.1 with its detailed proof in Appendix B. We are sincerely grateful for the insightful suggestions that helped us strengthen both the clarity and rigor of the paper, and we would be glad to address any further comments.

---

### Comment · Action_Editor_N3Y6 · 2025-11-18
**Another (short) delay**

Dear authors,

Thank you for your edits to the paper.

I am profoundly sorry for how long this decision is taking but there's a lot of discussion.

I will get back to you in the next few days.

Your AC

---

> ### Author Response · Authors · 2025-11-18
>
> Thank you for letting us know! We appreciate all the efforts and discussions from the AC and the reviewers that went into reviewing our paper. Please let us know if there is anything else we can do or any further clarification we can provide in the meantime. Looking forward to hearing from you soon!

---

### Decision · Action_Editor_N3Y6 · 2025-11-25

**Recommendation:** Accept with minor revision

**Additional Comments:**

There were two outstanding sources of criticism in the official recommendations.

The first was one reviewer's comments on the empirical evaluation; this was covered in the claims section above.

The second: two reviewers had legitimate outstanding questions surrounding the main theoretical contribution, Corrollary 5.1. This prompted a clarification and a request to update the paper, and the authors followed through.

The reviewers did confirm that the ambiguity around the "non-negligible" claim has been resolved by the updated paper, but brought up separate issues about how realistic the assumptions were. Specifically, the comments were:

(1)

> I have tried to evaluate the statement of the new results to the best of my abilities. The new version of Theorem 5.1 gives a lower bound on the probability that a sample will be generated, where this lower bound depends on the distance to the noise-free center of the model and how far off the manifold the sample is located, with the likelihood dropping rapidly as you go further off the manifold. Here, the authors have certainly addressed concerns about the theorem not being precise enough. However, there remains a gap between the theorem and the rest of the paper in that it remains unclear how much this tells us about the probability of generating out-of-combination (OOC) states. For this we would have to know whether OOC states lie along this manifold (in which case the theorem suggests they are likely to be generated) or far off the manifold (in which case the theorem suggests they are, in fact, unlikely to be generated). The authors address this by making the assumption that in-combinations states (both those in the training set and, crucially, OOC states) lie along a linear manifold. However, no arguments are given for why this assumption is reasonable, or why we should expect it to be true. I, therefore, cannot put too much stock in these results.

(2)

> I agree with reviewer **** that it is unclear whether the assumptions are meaningful. Assumption B.2 about the 'well-trained diffusion model' indeed does not seem to be in any way tied to the training distribution / training support. In fact it also has to hold outside the manifold of realistic data (the quantifier is over the ambient space).
>
> However, the authors do make clear that the OOC states are also on this linear manifold. However what was unclear  to me is where on the linear manifold they are. Eg my understanding is that because of the distribution shift, the model only sees some of the "orthants" on the manifold and the OOCs could be somewhere else.
>
> Finally I am somewhat surprised given the otherwise... more empirical focus of the original paper that there is now this math with no references or reuse of previous lemmas/theorems in the literature.
>
> Follow-up:
>
> Assumption B2, 1. The quantifier order is confusing, I would have expected the quantifier over x, y to be after the quantifier over t. This makes it sound like the constants can be dependent on x and y.  If it can depend also on them, the statement is trivial (the same as saying the denoiser should be bounded, if I'm not mistaken).
>
> Typos and suggestions:
>
> Eq 4 needs more bolding of x, y and
>
> - Clarify what is \cal{M}^{\bottom}
> - I think it's the orthogonal complement of the linear manifold.
> - "reserve mean" -> "reverse mean"?

Both reviewers admitted to not being experts in the area, so I recruited two more reviewers with experience in diffusion models. They did not submit full reviews, but participated in the discussion.

Their comments are:

(3)

> I have examined the work, with a specific focus on Theorem 5.1 (as highlighted). The underpinning hypothesis is, in my estimation, sound, and the accompanying proof appears to be mathematically sound. Nevertheless, I must reserve judgment on the practical significance of the established lower bound. While its non-zero nature is evident, I am currently unable to determine the scaling or magnitude of this quantity.

(4)

> Unfortunately I am also not an expert with the techniques used in Theorem 5.1. It sounds like other reviewers have determined that, with the modifications from the authors that include a rigorous formulation and proof in the appendix, the result was determined to be sound (at least to some extent). Rather than flagging the correctness of the result, I would instead flag the assumptions that are made on what constitutes a "well-trained diffusion model"-- the bi-Lipschitz criterion for example is definitely not standard in the literature of diffusion models.
>
> If this theorem is the only major concern remaining with the paper, I think that the burden should be on the authors to elaborate more on why these assumptions are reasonable, and how mild/strong they really are in practice. It is difficult to judge how valuable their mathematical result is otherwise. If other reviewers feel confident that the empirical aspects of the work are valuable and that the theorem is sound (in spite of having potentially too strong or impractical assumptions), it sounds reasonable to me to accept the work but strongly suggest to fix / conditionally accept / give authors another chance to address this.

These comments resolve my largest concern, which was the correctness of Corollary 5.1. However, it also does give two more opinions reinforcing the skepticism behind the practicality / applicability of the assumptions.

Ultimately, I think the purpose of this work is to propose a specific idea and that its merits are mainly concetrated on the empirical demonstration rather than the theoretical contribution. My read of sections 5 and 6 is that the authors are looking for a way to justify formally why diffusion models could be better at OOC. And, specifically, when considering TMLR's acceptance criteria, I do believe that the authors are upfront here and not over-claiming; they give one possible reason and treat it as such. However, the concerns surrounding the assumptions quoted above do need to be clarified, otherwise there is a mismatch in the narrative leading into the results that are used for the conclusions. E.g. are these assumptions even satisfied for the empirical evaluations that follow?

On a minor note, the approach reminds me of Unsupervised RL (at least the motivation for doing so is very similar) and I'm surprised to see no mention of it in the related work. I believe the related work should at least mention it in the RL subsection. Some relevant papers:

- Jaderberg et al. "Reinforcement Learning with Unsupervised Auxiliary Tasks", 2016.
- Ye et al. '25, "Implicit Search via Discrete Diffusion: A Study on Chess", 2025

but there may be more.

**Audience:**

Yes

**Audience Explanation:**

Absolutely -- no question here. Diffusion + RL both are relevant topics for TMLR.

**Claims And Evidence:**

Yes

**Claims Explanation:**

The paper introduces a new concept "out of combination" (OOC) generalization in the context of reinforcement learning. The main premise is that states/observations are typically a composition of many different components. Unseen states are combinations of the same set of components but just not specifically the combinations seen at train time.

The main claims are that training a diffusion model on successful trajectories generalizes better to unseen combinations than traditional RL methods. This is due to a conditional diffusion model that generates future states and actions, forcing it to understand / model / reproduce the dynamics of the world rather than to purely / only maximize reward.

The authors argue about why this approach will fair better both theoretically and demonstrate it across three domains in practice.

One of the three original reviewers in their original official recommendation still felt that the paper was too light on empirical demonstration (specifically that it was not compared to more imitation learning approaches), despite the authors adding a third domain. While this is true, it is a bit tangential to main topic of the paper which focuses on RL, and I think that the results are a good mix of zooming in to understand (Figure 3) and challenging domains (driving + multiagent Starcraft/SMAC). The diffusion approach comes out winning in every case, and at least in one case, it's demonstrated that it reduces error on OOC states. In my view, this is sufficient evidence to substantiate the claims.

The other remaining concern is whether the evidence supports the claims on the theoretical contribution (mainly Corollary 5.1). Ultimately, yes, they do.. but some some important clarifications need to be made. Please see the "Additional comments" for an extensive discussion on this point (and follow-up action items required by the authors).

---

> ### Author Response · Authors · 2025-11-27
>
> We sincerely thank the Action Editor and the reviewers for all the time and effort that went into reviewing our paper. We would also like to extend our appreciation to the external experts who provided invaluable feedback to our last draft. For our final revision, we have reorganized Section 5.2 and Section 6 so that Section 5.2 mainly includes the theoretical analysis while Section 6 focuses on the empirical considerations. Specifically, we discussed the (im)practicality of our assumptions and our model design choices to mitigate these issues in Section 6. We have also included a new discussion about unsupervised RL in the related works section. We hope this new (and final) manuscript provides a clearer connection between our theory and our experiments and addresses the concerns that the Action Editor and the reviewers had.
>
> This has been a wonderful peer-review experience for us, and our paper is significantly stronger after all the revisions. Thank you so much again for all the comments and feedback!